# Neural representation of newly instructed rule identities during early implementation trials

**Hannes Ruge[1]\*, Theo AJ Schäfer[1,2], Katharina Zwosta[1], Holger Mohr[1], Uta Wolfensteller[1]**

[1]Technische Universität Dresden, Dresden, Germany; [2]Max-Planck-Institute for Human Cognitive and Brain Sciences, Leipzig, Germany

**Abstract** By following explicit instructions, humans instantaneously get the hang of tasks they have never performed before. We used a specially calibrated multivariate analysis technique to uncover the elusive representational states during the first few implementations of arbitrary rules such as 'for coffee, press red button' following their first-time instruction. Distributed activity patterns within the ventrolateral prefrontal cortex (VLPFC) indicated the presence of neural representations specific of individual stimulus-response (S-R) rule identities, preferentially for conditions requiring the memorization of instructed S-R rules for correct performance. Identity-specific representations were detectable starting from the first implementation trial and continued to be present across early implementation trials. The increasingly fluent application of novel rule representations was channelled through increasing cooperation between VLPFC and anterior striatum. These findings inform representational theories on how the prefrontal cortex supports behavioral flexibility specifically by enabling the ad-hoc coding of newly instructed individual rule identities during their first-time implementation.

## Introduction

The prefrontal cortex (PFC) has been considered crucial for flexibly mastering the abundance of non-routine problems we are often facing (*Dosenbach et al., 2008*; *Duncan, 2001*; *Duncan, 2010*; *Miller and Cohen, 2001*; *Norman and Shallice, 1986*). The implementation of completely novel tasks for the very first time is a pivotal example of operating without routine solutions, as those are by definition unavailable (*Monsell, 1996*; *Norman and Shallice, 1986*). Moreover, it is essential to acquire novel tasks as rapidly as possible to ensure efficient performance and sometimes even physical integrity and ultimately survival. Humans are equipped with the highly developed ability of symbolic communication which is perfectly suited to acquire novel tasks in 'one shot' (*Greve et al., 2017*; *Lee et al., 2015*) simply by following explicit instructions. Thereby, more time-consuming and potentially costly trial-and-error learning can be avoided (*Doll et al., 2009*; *Noelle, 1997*; *Ruge et al., 2018a*).

Earlier research has generated first insights into the neural basis of 'instruction-based learning' or 'rapid instructed task learning' (*Cole et al., 2017*; *Liefooghe et al., 2018*; *Wolfensteller and Ruge, 2012*). However, it has remained elusive whether and, if so, how the concrete rules of newly instructed tasks are initially represented in the human PFC during early implementation trials right after their first-time instruction. By addressing these questions, the present study set out to inform representational theories of PFC functioning regarding the type and the timescale of task-related information coded within PFC regions. Specifically, we sought to identify distributed neural activity patterns associated with subtle representational differences regarding newly instructed individual rule identities such as 'if the word BUTTER is displayed on the screen, then flex the middle finger' or

\*For correspondence: hannes.ruge@tu-dresden.de

**Competing interests:** The authors declare that no competing interests exist.

'if the word MONKEY is displayed on the screen, then flex the index finger'. To this end, we employed a recently developed multivariate pattern analysis technique (MVPA) specifically calibrated to uncover the rapidly evolving representational dynamics while implementing novel rule instructions for the first time (*Ruge et al., 2018b*). Importantly, this technique (see Materials and methods) ensured unbiased results by avoiding systematic imbalance in model regressor correlations through appropriate stimulus sequence construction (cf., *Mumford et al., 2014*; *Visser et al., 2016*).

Tracking these fine-grained representational dynamics is crucial for a comprehensive understanding of the rapid neural re-organization processes that are taking place during early implementation trials right after first-time task instruction. Such rapid neural re-organization processes have been evidenced in terms of both mean activity dynamics (*Cole et al., 2010*; *Hartstra et al., 2011*; *Ruge and Wolfensteller, 2010*; *Sheffield et al., 2018*) and connectivity dynamics (*Hampshire et al., 2019*; *Mohr et al., 2016*; *Mohr et al., 2018*; *Ruge and Wolfensteller, 2013*). Specifically, conventional univariate analysis of *mean activity* has shown that lateral PFC engagement was maximal during the first-time implementation, followed by a rapid decline across the first few implementation trials (*Hampshire et al., 2019*; *Hartstra et al., 2011*; *Ruge and Wolfensteller, 2010*). This was paralleled by increasing fronto-striatal *functional connectivity* (*Ruge and Wolfensteller, 2013*; *Ruge and Wolfensteller, 2015*). Together, these earlier observations suggested that short-term task automatization processes enable increasingly fluent task implementation by support of increasing inter-regional cooperation (*Chein and Schneider, 2012*; *Mohr et al., 2016*; *Ruge and Wolfensteller, 2016*).

Crucially, however, based on general methodological considerations (*Coutanche, 2013*), mean activity and connectivity dynamics are uninformative regarding the rule-specific *representational* dynamics being expressed in spatially distributed activity patterns. These can only be uncovered via time-resolved MVPA and by testing a number of alternative scenarios. To start with, representations of newly instructed task rules might be detectable within prefrontal cortex as early as in the first implementation trial right after their first-time instruction. If so, the next question then regards the continued presence of such type of representation. One possibility is that the initial presence of prefrontal rule representations is rapidly fading at the same pace as cognitive control requirements are decreasing (as evidenced by rapidly decreasing mean PFC activity). Alternatively, the presence of prefrontal rule representations might continue to be important for successful task implementation as their more and more fluent application is increasingly channeled through fronto-striatal inter-regional cooperation. A radically different possibility is that novel prefrontal task representations are not yet in place immediately after instruction but are instead being built over the first few implementation trials again based on increasing fronto-striatal cooperation but this time with a leading role of striatal areas. This would be consistent with results in non-human primates during trial-and-error learning showing that successful rule acquisition occurred a few trials *before* rule-specific neural coding could be detected in the lateral PFC (*Cromer et al., 2011*; *Pasupathy and Miller, 2005*).

Importantly, if any of these scenarios could be confirmed empirically, this would provide first evidence for human PFC representing entirely novel task rules in the initial phase of task implementation. This contrasts with existing MVPA studies, which have shown that prefrontal cortex regions flexibly code currently task-relevant information, but these studies involved already *well-familiarized* tasks (*Jackson and Woolgar, 2018*; *Woolgar et al., 2015*). A few pioneering studies have shown that such prefrontal representations are retrieved and re-cycled in the service of newly instructed tasks that rely on, or are recomposed of familiar task elements such as familiar semantic judgments or familiar perceptual categorizations (*Cole et al., 2011*; *Cole et al., 2013*; *Muhle-Karbe et al., 2017*). Yet, these studies were not designed to determine how the newly recomposed rule identities were individually represented in the brain.

We conducted two inter-related fMRI experiment both involving a large number of different learning blocks each comprising a new and unique set of instructed stimulus-response (S-R) rules (see *Figures 1* and *2*). MVPA was used to identify activity patterns sensitive to individual stimulus-response rule identities across the first few implementation trials following novel instructions (see *Figure 3*). This was done separately for each task block before aggregation across blocks.

Besides the primary goal to examine rule-specific representational dynamics, experiment 1 was additionally designed to explore the relationship between the strength or integrity of prefrontal rule representations and the commission of performance errors. To this end, the proportion of performance errors was manipulating by varying the complexity or difficulty of S-R instructions. If performance errors were due to compromised integrity of S-R rule representations, a higher percentage of

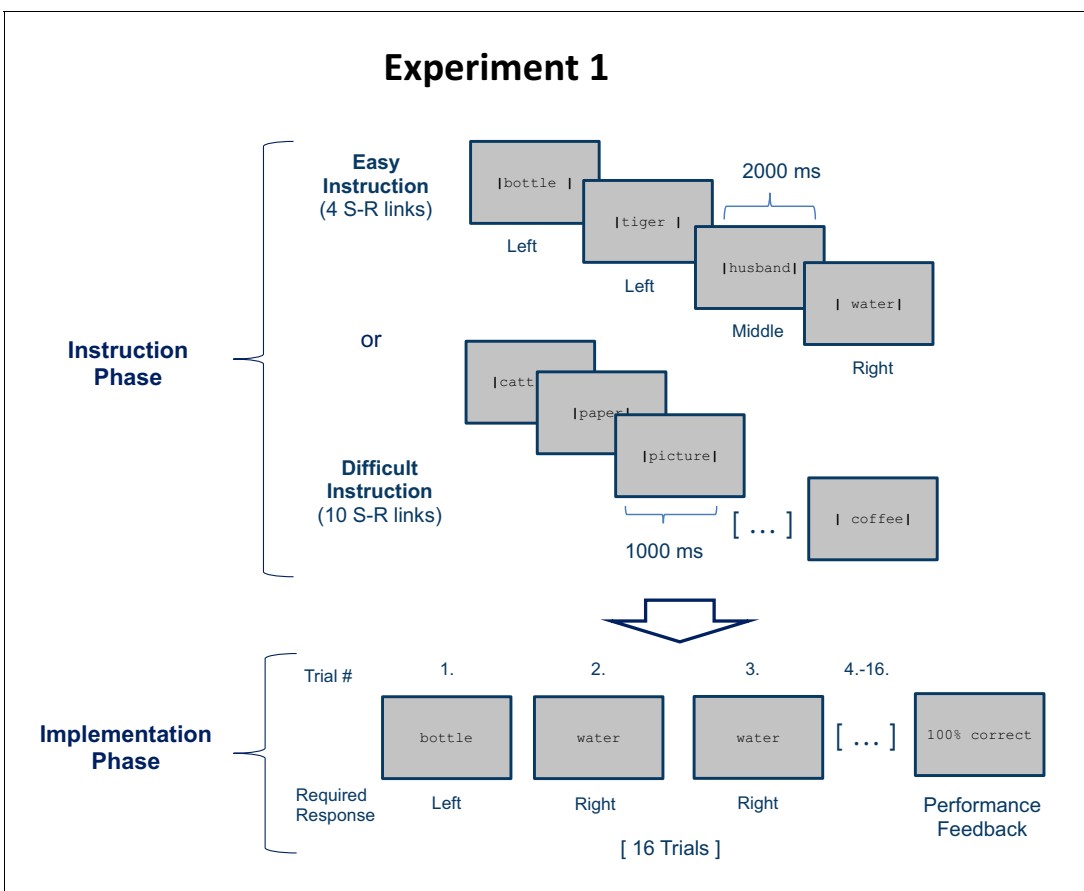

**Figure 1.** Stimulus-response (S–R) learning task used in experiment 1 exemplarily depicted for one of 18 blocks per condition (easy and difficult). Each block consisted of an instruction phase and an implementation phase. During the instruction phase participants were presented with 4 (easy instruction) or 10 (difficult instruction) pairings between disyllabic nouns and manual responses. The vertical bars framing the nouns indicated the correct response (e.g. Bottle - left). During the subsequent implementation phase (here, selectively shown for the easy condition), each nouns was presented 4 times in random order without the vertical bars and participants had to respond as instructed. Irrespective of S-R rule difficulty (4 vs. 10 nouns in the instruction phase), a constant number of 4 different nouns was presented in the implementation phase. At the end of each block, feedback specifying the percentage of correctly answered trials was displayed.

error trials included in the MVPA following more difficult instructions should imply weaker rule-specific activity patterns (*Cole et al., 2016*; *Rigotti et al., 2013*). Alternatively, according to the notion of 'goal neglect' asserting that 'knowing' is not necessarily the same as 'doing' (*Duncan et al., 1996*; *Duncan et al., 2008*), more complex instructions might induce more errors despite largely intact prefrontal rule representations. Instead, more complex instructions might absorb control resources that are then missing to prevent competing (e.g., perseverative) response tendencies from overriding the instructed response. In this case, rule-specific activity patterns should remain unaffected by error rate differences induced by more or less complex instructions.

Experiment 2 was designed as a follow-up to experiment 1 to specifically test the hypothesis that prefrontal cortex representations can be identified preferentially for *intentional* learning conditions involving newly instructed stimulus-response rules which were to be memorized for correct performance later on. To this end, the intentional learning condition was compared to a control condition involving the same contingencies between the same stimuli and responses, but without the necessity to memorize these contingencies for correct performance. Specifically, novel stimuli were presented together with additional cues denoting the correct response throughout the entire implementation phase in the control condition of experiment 2. Hence, correct performance was perfectly possible without memorization of the newly introduced S-R contingencies. This experimental rationale was based on a recent behavioral study which suggested that subjects refrain from intentional S-R learning in the absence of obligatory memorization demands (*Ruge et al., 2018a*). Specifically, response

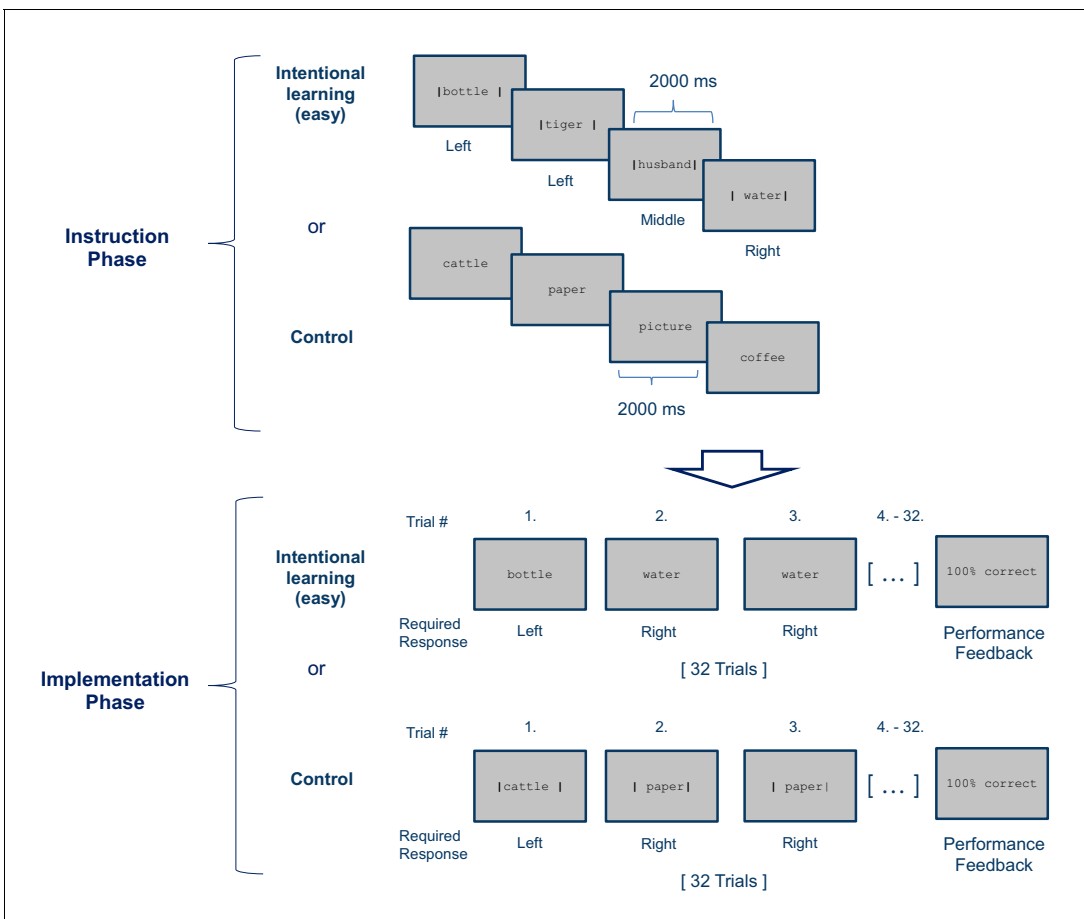

**Figure 2.** Stimulus-response (S–R) learning task used in Experiment 2 exemplarily depicted for one of 12 blocks per condition (Intentional learning vs. control). As in experiment 1, each block consisted of an instruction phase and an implementation phase. The Intentional learning condition was identical to the easy condition of experiment 1 (i.e., four instructed S-R rules) except that each S-R rule needed to be implemented 8 times instead of 4 times. In the control condition the response cues (i.e. the vertical bars) were omitted during the instruction phase and were instead presented during the subsequent implementation phase. At the end of each block, feedback specifying the percentage of correctly answered trials was displayed.

cues presented during the first few novel implementation trials were processed very differently depending on their continued vs. discontinued presence in later implementation trials. When subjects knew that response cue presentation discontinued (i.e. S-R memorization was required for correct performance later on), they spent additional effort into encoding the instructed S-R rules and this encoding effort predicted subsequent memory-based performance. All of these effects were absent when subjects knew that response cue presentation would continue indefinitely (as in the present control condition). Applied to the present study this implies that the control condition of experiment 2 likely does *not* involve intentional S-R learning processes.

## Results

### Behavioral performance (experiment 1)

RTs and response accuracies from experiment 1 were analyzed with repeated measures ANOVAs. Each ANOVA included the independent variables stimulus repetition (with the levels 1 to 4) and instruction difficulty (with the levels easy and difficult). Greenhouse-Geisser correction was applied where necessary. The results are visualized in *Figure 4*.

The ANOVA for RTs revealed a significant RT decrease across stimulus repetitions ($F_{3,192}=224.87$; $p(F)<0.001$; $\eta_p^2=0.78$; linear contrast: $F_{1,64}=290.67$ $p(F)<0.001$; $\eta_p^2=0.82$) which was more pronounced for difficult compared to easy instruction blocks ($F_{3,192}=137.94$; $p(F)<0.001$; $\eta_p^2=0.68$; linear

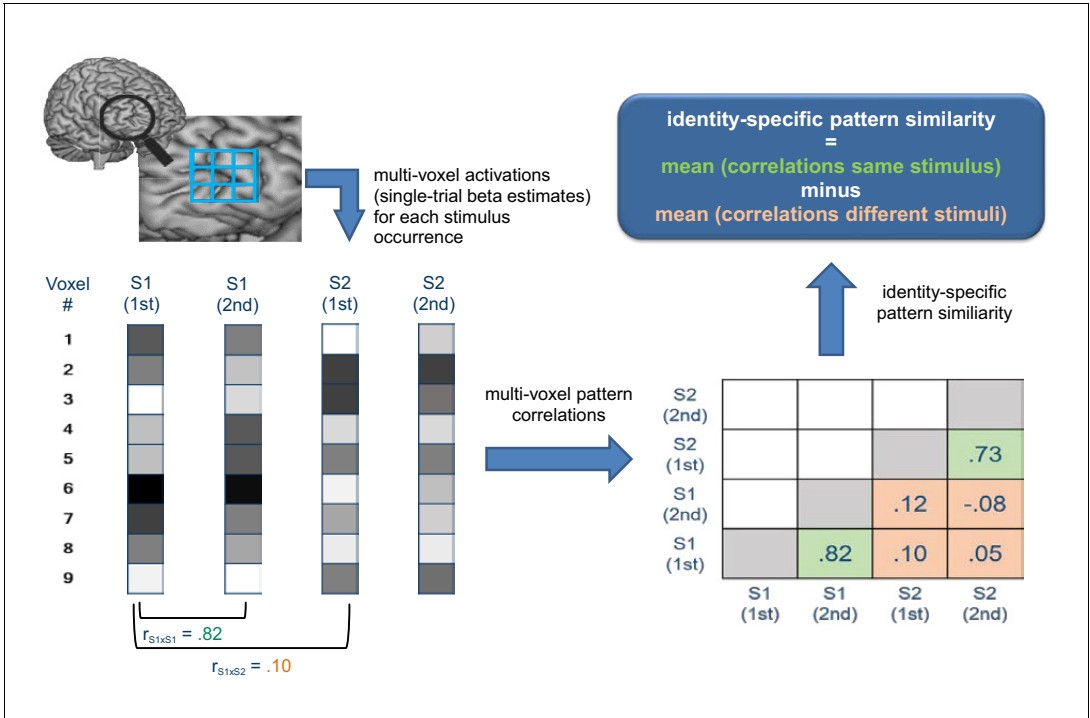

**Figure 3.** Schematic illustration depicting how identity-specific multi-voxel pattern similarity was computed exemplarily for one implementation stage in one learning block. For illustrative purposes, only two stimuli (S1 and S2) each occurring twice are considered here (instead of 4 stimuli in reality). Bottom left: For each stimulus occurrence voxel-wise beta estimates (visualized by grayscale values) are arranged in vectors that constitute the basis of multi-voxel pattern correlations. Bottom right: matrix values depict multi-voxel pattern correlations for all combinations of trials. Green cells denote correlations between same stimuli, orange cells denote correlations between different stimuli. Top right: Identity-specific pattern similarity is defined by significantly greater mean correlations in green cells compared to orange cells.

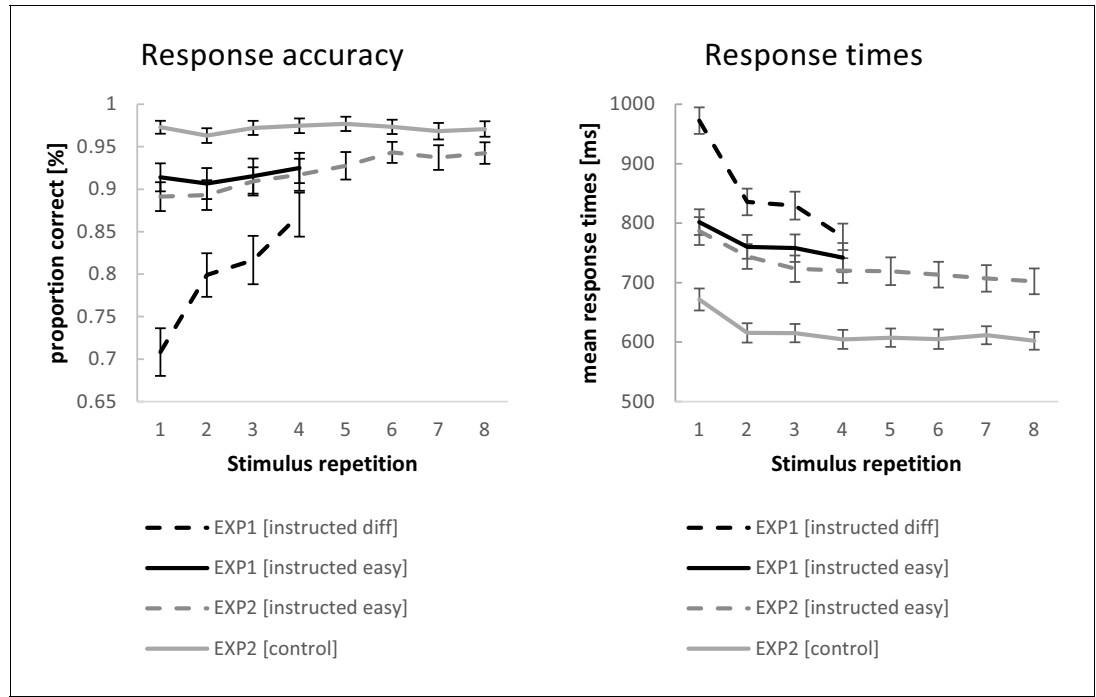

**Figure 4.** Behavioral performance data for experiment 1 and experiment 2. Error bars represent 90% confidence intervals.

contrast: $F_{1,64}$=252.69; p(F)<0.001; $\eta_p^2$=0.80) on top of generally slower RTs ($F_{1,64}$=175.15; p(F) <0.001; $\eta_p^2$=0.73). Even at stimulus repetition 4, RTs were still significantly slower for difficult blocks relative to easy blocks (t = 4.60; p(t)<0.001).

The ANOVA for response accuracies revealed a significant increase in accuracies across stimulus repetitions ($F_{3,192}$=71.43; p(F)<0.001; $\eta_p^2$=0.53; linear contrast: $F_{1,64}$=111.52 p(F)<0.001; $\eta_p^2$=0.64). This increase was more pronounced for difficult compared to easy instruction blocks ($F_{3,192}$=80.14; p (F)<0.001; $\eta_p^2$=0.56; linear contrast: $F_{1,64}$=156.40; p(F)<0.001; $\eta_p^2$=0.71) on top of generally higher accuracies for easy blocks than difficult blocks ($F_{1,64}$=202.81; p(F)<0.001; $\eta_p^2$=0.76). Even at stimulus repetition 4, accuracies were still significantly higher for easy blocks relative to difficult blocks (t = 6.26; p(t)<0.001).

Accuracy was positively correlated with the progressive matrices intelligence score, both for easy instructions (r = 0.32; p=0.005 one-tailed) as well as for difficult instructions (r = 0.35; p=0.002 one-tailed). The correlation between the intelligence score and the accuracy difference between easy and difficult instructions showed a trend towards significance (r = -.19; p=0.066 one-tailed) indicating that more intelligent participants suffered less from more difficult instructions relative to the easier instructions. Analogous correlations between accuracies and forward and backward simple digit span scores were all non-significant (all p(r)>0.14).

## Behavioral performance (experiment 2)

The behavioral data from experiment two were analyzed with repeated measures ANOVAs including the independent variables stimulus repetition (with the levels 1 to 8) and instruction type (with the levels intentional learning and control). Greenhouse-Geisser correction was applied where necessary. The results are visualized in *Figure 4*. Entering mean response times (RT) as the dependent variable revealed a significant RT decrease across stimulus repetitions ($F_{7,483}$=42.00; p(F)<0.001; $\eta_p^2$=0.38; linear contrast: $F_{1,69}$=79.25 p(F)<0.001; $\eta_p^2$=0.54), which was more pronounced for intentional learning blocks compared to control blocks ($F_{7,483}$=4.40; p(F)=0.002; $\eta_p^2$=0.06; linear contrast: $F_{1,69}$=6.93; p(F)=0.010; $\eta_p^2$=0.09) on top of generally slower RTs ($F_{1,69}$=181.17; p(F)<0.001; $\eta_p^2$=0.72). At stimulus repetition 8, RTs were still significantly slower for intentional learning blocks relative to control blocks (t = 12.34; p(t)<0.001).

Entering response accuracies as the dependent variable revealed a significant increase in accuracies across stimulus repetitions ($F_{7,483}$=17.51; p(F)<0.001; $\eta_p^2$=0.20; linear contrast: $F_{1,69}$=45.88; p(F) <0.001; $\eta_p^2$=0.40). This increase was more pronounced for intentional learning compared to control blocks ($F_{7,483}$=13.95; p(F)<0.001; $\eta_p^2$=0.17; linear contrast: $F_{1,69}$=49.10; p(F)<0.001; $\eta_p^2$=0.42) on top of generally higher accuracies for control blocks than intentional learning blocks ($F_{1,69}$=77.18; p (F)<0.001; $\eta_p^2$=0.53). At stimulus repetition 8, accuracies were still significantly higher for control blocks relative to intentional learning blocks (t = 5.40; p(t)<0.001).

## MVPA (experiment 1)

ROI-based estimates of identity-specific activation patterns from experiment 1 were submitted to a 4-factorial repeated-measures ANOVA including the independent variables implementation stage (early vs. late) and difficulty (easy vs. diff) and additionally region (VLPFC vs. DLPFC) and hemisphere (left vs. right) in order to adequately account for potential regional differences (*Nieuwenhuis et al., 2011*). The results are visualized in *Figure 5*. This analysis yielded a significant overall identity-specific pattern similarity effect defined by the mean across all conditions (constant term: $F_{1,64}$=7.9; p(F) =0.006; $\eta_p^2$=0.11) and a significantly stronger pattern similarity effect for VLPFC than DLPFC (main effect region: $F_{1,64}$=13.1; p(F)<0.001; $\eta_p^2$=0.17).

There were no significant effects involving implementation stage but a trend towards a smaller effect for late vs. early in the VLPFC compared to the DLPFC (interaction stage by region: $F_{1,64}$=3.6; p(F)=0.064; $\eta_p^2$=0.053). In order to test whether this trend might point towards a 'true' but small effect that was missed due to insufficient statistical power, we conducted an additional more powerful analysis by collapsing data across experiments 1 and 2. However, this analysis again did not produce reliable evidence for a significant influence of implementation stage (for details see further below). Also, there were no significant effects involving difficulty. If anything, contrary to the prediction of weakened rule representations, there was a trend towards a *stronger* identity-specific pattern similarity effect in the difficult condition compared to the easy condition (main effect difficulty:

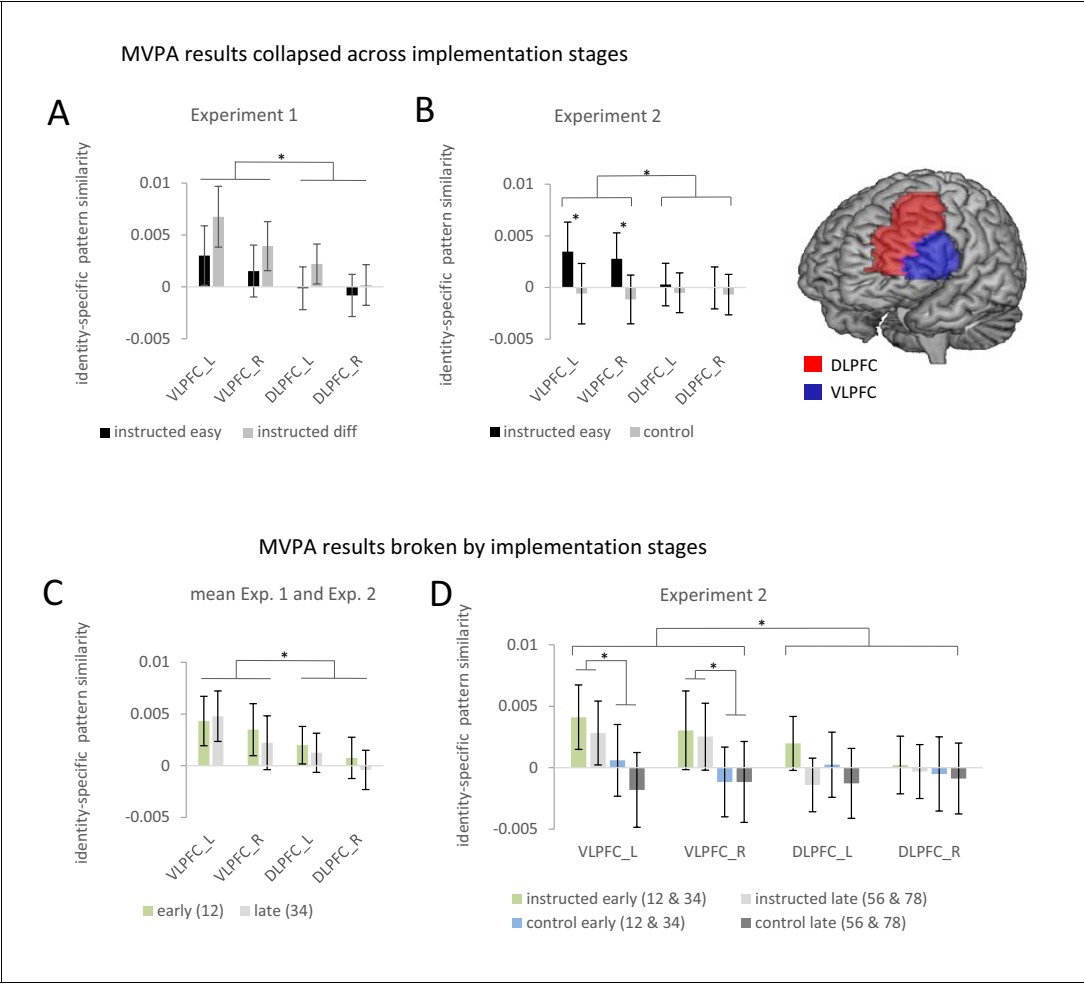

**Figure 5.** Summary of the ROI-based MVPA results for experiments 1 and 2. Error bars represent 90% confidence intervals. Significant differences are indicated by asterisks. (**A**) Identity-specific pattern similarities in experiment 1 collapsed across implementation stages. (**B**) Identity-specific pattern similarities in experiment 2 collapsed across implementation stages. (**C**) Identity-specific pattern similarities collapsed across experiments 1 and 2 broken by implementation stages. Early implementation stage pattern similarities are based on stimulus repetitions 1 and 2 whereas late implementation stage pattern similarities are based on stimulus repetitions 3 and 4. (**D**) Identity-specific pattern similarities for experiment 2 broken by implementation stages. Early implementation stage pattern similarities are based on aggregated values for stimulus repetitions 1 and 2 and stimulus repetitions 3 and 4. Late implementation stage pattern similarities are based on aggregated values for stimulus repetitions 3 and 4 and stimulus repetitions 7 and 8.

$F_{1,64}=3.3$; p(F)=0.074; $\eta_p^2=0.049$). This trend towards stronger identity-specific rule representation in difficult blocks might hint towards a sharper representation of rules specifically in correctly performed difficult trials compared to correctly performed easy trials which might have been blurred by inclusion of erroneous trials (cf., *Woolgar et al., 2015*). If this were true, MVPA comprising correct trials only, should boost the trend for stronger effects in difficult blocks compared to easy blocks. We therefore repeated the ROI-based MVPA including correct trials only. However, the results remained qualitatively the same. If anything the trend towards a stronger effect in the difficult condition became weaker (main effect difficulty: $F_{1,64}=1.12$; p(F)=0.294; $\eta p2 = 0.017$).

The ROI-based findings were confirmed by searchlight-based MVPAs within each ROI, revealing a significant overall identity-specific pattern similarity effect defined by the mean across all conditions specifically within the left VLPFC (MNI: −48 5 23; t = 5.09; $p_{FWE}$ <0.001 and MNI: −45 32 11; t = 4.66; $p_{FWE}$ <0.001) and a trend in the same direction within the right VLPFC (MNI: 48 8 11; t = 3.16; $p_{FWE}$ <0.074). Again, there were no significant effects involving implementation stage or difficulty.

On the whole brain level, the searchlight MVPA confirmed for the left VLPFC that the overall identity-specific pattern similarity effect defined by the mean across all conditions was significant even after correction for the whole-brain volume (MNI: −48 5 23; t = 5.09; $p_{FWE}$ = 0.005 and MNI: −45 32 11; t = 4.66; $p_{FWE}$ = 0.028). Additionally, this analysis revealed significant whole-brain-corrected overall identity-specific pattern similarity effects in the left sensorimotor cortex (MNI: −39–25 53; t = 9.28; $p_{FWE}$ <0.001) and in the left visual cortex (MNI: −15–91 −7; t = 6.52; $p_{FWE}$ <0.001). There were no significant effects involving implementation stage or difficulty. These findings are as expected and consistent with the coding of stimulus identity in the visual cortex and response identity in the left sensorimotor cortex, respectively (see *Figure 6*).

## MVPA (experiment 2)

Instead of reflecting S-R rule-specific representations, the findings of experiment 1 could in principle reflect representations of stimulus identity or response identity alone. In fact, this was very likely the case for the left sensorimotor cortex (response identity) and for the visual cortex (stimulus identity). To clarify this, experiment 2 included a control condition which was identical in terms of information content regarding word stimuli, responses, and contingencies between words and responses. The only difference was that novel S-R rules were not required to be actively memorized for correct performance in present or future trials, and hence intentional encoding was discouraged. This was done by presenting explicit response cues on every trial together with the word stimuli throughout the entire implementation phase. Since subjects were aware of this, we reasoned that they would not intentionally encode and later retrieve S-R rules in working memory (*Cole et al., 2017*) or episodic long-term memory (*Meiran et al., 2017*; *Ruge et al., 2018a*). Evidence for this claim comes from behavioral results obtained using a similar manipulation in a recent purely behavioral study (*Ruge et al., 2018a*).

ROI-based estimates of identity-specific activation patterns were submitted to a 4-factorial repeated-measures ANOVA including the independent variables implementation stage (early vs. late), instruction type (intentional learning vs. control), and additionally region (VLPFC vs. DLPFC) and hemisphere (left vs. right) in order to adequately account for potential regional differences (*Nieuwenhuis et al., 2011*). The results are visualized in *Figure 5*. Note that different from experiment 1, this time the early implementation stage comprised the mean across identity-specific pattern similarities computed for stimulus repetitions 1 and 2 and stimulus repetitions 3 and 4, respectively. The late implementation stage comprised the mean across identity-specific pattern similarities computed for stimulus repetitions 5 and 6 and stimulus repetitions 7 and 8, respectively. The ANOVA yielded a significantly stronger pattern similarity effect in the intentional learning condition than in the control condition (main effect of instruction type: $F_{1,69}$=4.49; p(F)=0.038; $\eta_p^2$=0.061) and this difference was significantly stronger for VLPFC than DLPFC (interaction instruction type by region: $F_{1,69}$=6.91; p(F)<0.011; $\eta_p^2$=0.091). Post-hoc tests showed that the main effect of instruction type was exclusively significant in the VLPFC both in the early implementation stage ($F_{1,69}$=4.60; p(F) =0.036; $\eta_p^2$=0.062) and in the late implementation stage ($F_{1,69}$=4.05; p(F)=0.048; $\eta_p^2$=0.055), but not in the DLPFC (early: $F_{1,69}$=.46; p(F)=0.50; $\eta_p^2$=0.007; late: $F_{1,69}$=.012; p(F)=0.91; $\eta_p^2$=0.0001). There was no significant effect involving implementation stage. Unlike experiment 1, there was not even a trend towards an influence of implementation stage when testing the relevant interaction involving stage, region, and instruction type ($F_{1,69}$=.22; p(F)=0.641; $\eta_p^2$=0.003). Note that similar results were obtained when implementation stage comprised all four non-aggregated levels (i.e. based on stimulus repetitions 1/2, 3/4, 5/6, and 7/8) instead of the two aggregated levels used in the primary analysis.

Thus, the ROI-based MVPA confirmed the findings from experiment 1 and importantly showed that identity-specific MPVA effects are indeed specific of intentional S-R learning conditions as compared to the control condition involving the same stimuli and responses. Notably, again consistent with experiment 1, the MVPA effects for intentional S-R learning relative to control were significantly stronger in the VLPFC compared to the DLPFC where an effect was virtually absent.

These ROI-based findings were confirmed by searchlight-based MVPAs within each ROI revealing stronger identity-specific pattern similarity effects in the instructed condition than in the control condition specifically within the left VLPFC ROI (MNI: −36 17 26; t = 3.51; $p_{FWE}$ = 0.025) and a trend in the same direction also within the right VLPFC (MNI: 60 14 14; t = 2.90; $p_{FWE}$ = 0.134). On the whole brain level, the searchlight MVPA did not reveal additional regions exhibiting a main effect of

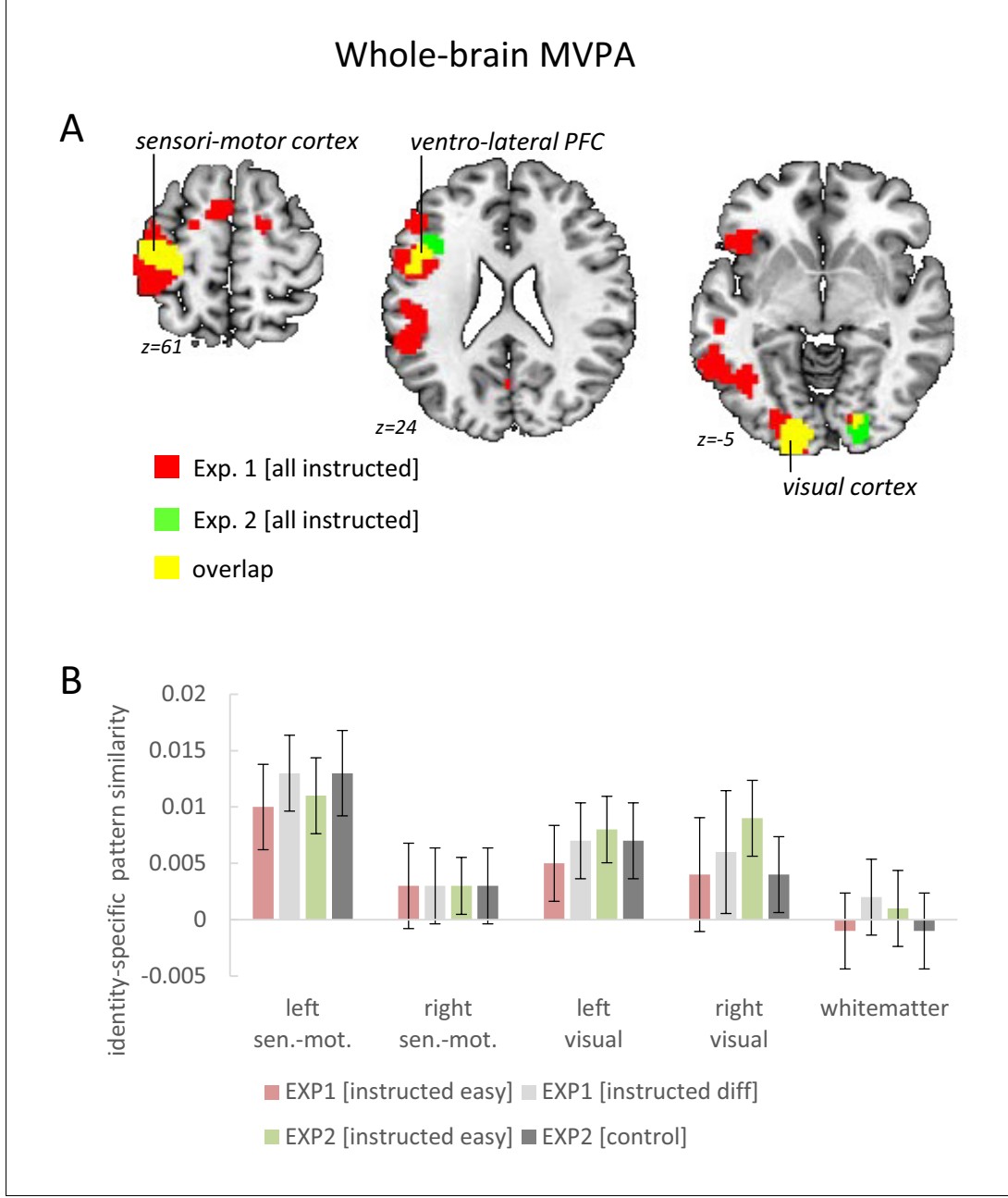

**Figure 6.** Results of the whole-brain searchlight MVPA testing for overall identity-specific pattern similarity effects. (A) Horizontal brain slices depicting the findings for the left sensorimotor cortex, the ventro-lateral PFC, and the visual cortex. For display purposes the map shows voxels with p<0.001 uncorrected. (B) Pattern-similarity effects broken by instruction difficulty (exp. 1) or instruction type (exp. 2). In addition to sensorimotor cortices and visual cortices, the white-matter volume is included as a control region to highlight the absence of analysis bias. For a comprehensive summary of ventro-lateral PFC results see **Figure 5**. Error bars represent 90% confidence intervals.

instruction type. Neither implementation stage nor instruction-type had a significant influence on the whole-brain searchlight results. Notably, testing for identity-specific activation patterns collapsed across intentional learning blocks and control blocks revealed the expected effects for both conditions alike in the sensorimotor cortex (MNI: −39–25 50; t = 10.48; $p_{FWE}$ <0.0001) and the visual cortex (MNI: −15–91 −4; t = 5.1; $p_{FWE}$ = 0.003 and MNI: 21–88 −4; t = 4.67; $p_{FWE}$ = 0.02). These results are depicted in **Figure 6** and confirm the findings from experiment 1. Importantly, different from the VLPFC findings which were highly specific for the intentional learning condition, visual

cortex and sensorimotor cortex exhibited – as expected – comparable effects both in the intentional learning condition as well as in the control condition. This is consistent with representations of stimulus identity and response identity, respectively.

## MVPA (collapsed across experiments 1 and 2)

Experiment 1 exhibited a non-significant trend towards weaker identity-specific pattern similarity for the late implementation stage relative to the early implementation stage. In order to test whether this trend might point towards a 'true' but small effect that was missed due to insufficient statistical power, we conducted an additional more powerful analysis based on data from both experiments. Data from experiments 1 and 2 were jointly analyzed including all the intentional learning conditions (i.e., omitting the control condition from experiment 2) for the early stage spanning stimulus repetitions 1 and 2 and the late stage spanning stimulus repetitions 3 and 4 (i.e., omitting stimulus repetitions 5/6 and 7/8 from experiment 2). The results are visualized in *Figure 5C*.

ROI-based estimates of identity-specific pattern similarity were submitted to a 3-factorial repeated-measures ANOVA including the independent variables implementation stage (early vs. late), region (VLPFC vs. DLPFC), and hemisphere (left vs. right). Not surprisingly, this ANOVA again yielded a significant overall identity-specific pattern similarity effect defined by the mean across all conditions (constant term: $F_{1,134}=9.37$; p(F)=0.003; $\eta_p^2=0.065$) and a significantly stronger pattern similarity effect for the VLPFC than the DLPFC (main effect region: $F_{1,134}=14.04$; p(F)<0.001; $\eta_p^2=0.095$). Most importantly, refuting the preliminary trend observed in experiment 1, this latter effect was *not* significantly affected by implementation stage (interaction stage by region: $F_{1,134}=.47$; p(F)=0.49; $\eta_p^2=0.004$). All other ANOVA effects involving implementation stage were also non-significant (all p>0.40). Finally, to establish that an identity-specific pattern similarity effect was present in each implementation stage two additional ANOVAs were computed each restricted to a single stage (i.e., early: repetitions 1/2 and late: repetitions 3/4). Specifically, for the early stage, there was a significant main effect region ($F_{1,134}=5.61$; p(F)=0.019; $\eta_p^2=0.04$) reflecting a significant effect for the VLPFC ($F_{1,134}=9.60$; p(F)=0.002; $\eta_p^2=0.067$) but not for the DLPFC ($F_{1,134}=1.78$; p(F)=0.184; $\eta_p^2=0.013$). For the late stage, there was again a significant main effect region ($F_{1,134}=9.41$; p(F)=0.003; $\eta_p^2=0.066$) reflecting a significant effect for the VLPFC ($F_{1,134}=7.53$; p(F)=0.007; $\eta_p^2=0.053$) but not for the DLPFC ($F_{1,134}=.001$; p(F)=0.974; $\eta_p^2=0.001$). Hence, overall, it seems relatively safe to conclude that identity-specific pattern similarity effects in the VLPFC were present across all implementation stages.

## Univariate analysis (experiment 1)

A complementary ROI-based univariate analysis for experiment 1 was based on condition-specific mean activity estimates which were submitted to a 4-factorial repeated-measures ANOVA including the independent variables stimulus repetition (1 to 4), difficulty (easy vs. diff), region (VLPFC vs. DLPFC), and hemisphere (left vs. right). The results are visualized in *Figure 7*. There was a significant main effect of stimulus repetition ($F_{3,192}=62.38$; p(F)<0.001; $\eta_p^2=0.29$) reflecting a general linear activation decrease (linear contrast: $F_{1,64}=187.11$; p(F)<0.001; $\eta_p^2=0.39$). A significant three-way interaction involving difficulty, region, and hemisphere ($F_{1,64}=7.20$; p(F)=0.009; $\eta_p^2=0.10$) reflected stronger activation in the difficult condition relative to the easy condition which was especially pronounced in the left DLPFC. This was further qualified by a significant four-way interaction additionally including stimulus repetition ($F_{3,192}=5.96$; p(F)=0.001; $\eta_p^2=0.09$) reflecting a linearly decreasing influence of difficulty which was especially pronounced in the left VLPFC (linear contrast: $F_{1,64}=9.82$; p(F)=0.003; $\eta_p^2=0.13$).

## Univariate analysis (experiment 2)

A complementary ROI-based univariate analysis for experiment 2 was based on condition-specific mean activity estimates which were submitted to a 4-factorial repeated-measures ANOVA including the independent variables stimulus repetition (1 to 8), instruction type (intentional learning vs. control), region (VLPFC vs. DLPFC), and hemisphere (left vs. right). The results are visualized in *Figure 7*. This ANOVA yielded significant main effects of instruction type ($F_{1,69}=5.31$; p(F)=0.024; $\eta_p^2=0.071$) and stimulus repetition ($F_{7,483}=14.51$; p(F)<0.001; $\eta_p^2=0.174$) indicating generally higher activation for the intentional learning blocks relative to the control blocks and generally decreasing activation

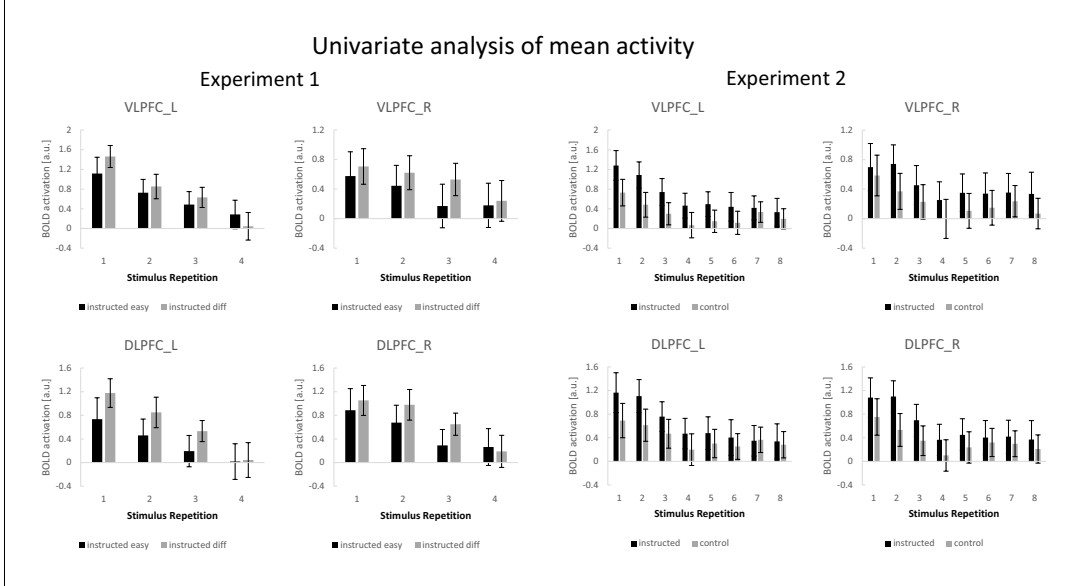

**Figure 7.** Summary of the ROI-based mean activity results for experiments 1 and 2. Error bars represent 90% confidence intervals.

across stimulus repetitions. Notably, however, a significant four-way interaction between all independent variables ($F_{7,483}$=4.27; p(F)=0.002; $\eta_p^2$=0.058) indicated that the stronger activation for intentional learning blocks was linearly decreasing across stimulus repetitions, but to a different extent across ROIs and particularly pronounced for the left VLPFC (linear contrast: $F_{1,69}$=10.54; p(F)=0.002; $\eta_p^2$=0.133).

## Functional connectivity analysis (experiment 2)

Previous studies have reported *increasing* connectivity across stimulus repetitions between the LPFC and the anterior striatum under instruction-based learning conditions (*Ruge and Wolfensteller, 2013*; *Ruge and Wolfensteller, 2015*). The study design of the present experiment 2 offers the unique opportunity to explicitly test whether this effect is specific of intentional learning blocks compared to control blocks. Such a finding would additionally inform the MVPA results by suggesting that the repeated application of newly established VLPFC rule representations might be increasingly channelled through inter-regional cooperation between the VLPFC and the anterior striatum.

Analogously to the earlier studies, we tested for a stronger functional connectivity increase from early implementation trials (stimulus repetitions 1 and 2) to late implementation trials (stimulus repetitions 7 and 8). The results are visualized in *Figure 8*. Using the left VLPFC as seed region, we specifically tested for significant beta-series correlation effects within an anatomically defined basal ganglia ROI comprising all of caudate nucleus, putamen, and pallidum. This revealed the predicted effect in the anterior striatum (MNI: −6 14–4; t = 4.12; p(t)=0.016 and MNI: 6 14–7; t = 4.17; p(t) =0.014; FWE-corrected for the basal ganglia volume). There were no additional regions identified after correction for the whole brain volume. Note that also the striatal activation dynamics were as expected based on previous studies. Specifically, as visualized in *Figure 8C*, there was a significant linear activation increase during intentional learning blocks relative to control blocks for the anterior striatum cluster identified in the connectivity analysis (MNI: 6 14–7; t = 4.35; p(t)=0.001 and MNI: −9 20–7; t = 4.53; p(t)=0.001 and MNI: 18 23–7; t = 4.96; p(t)<0.001; all FWE-corrected for the ant. striatum volume). Together, these findings lend further support for an early practice-related increase in anterior striatal activity and connectivity specifically under instruction-based learning conditions as has been debated recently (*Hampshire et al., 2019*; *Ruge and Wolfensteller, 2016*).

## Discussion

The key finding from our time-resolved MVPA is that rule identity-specific representations were detectable starting from the first implementation trial immediately after the first-time instruction of

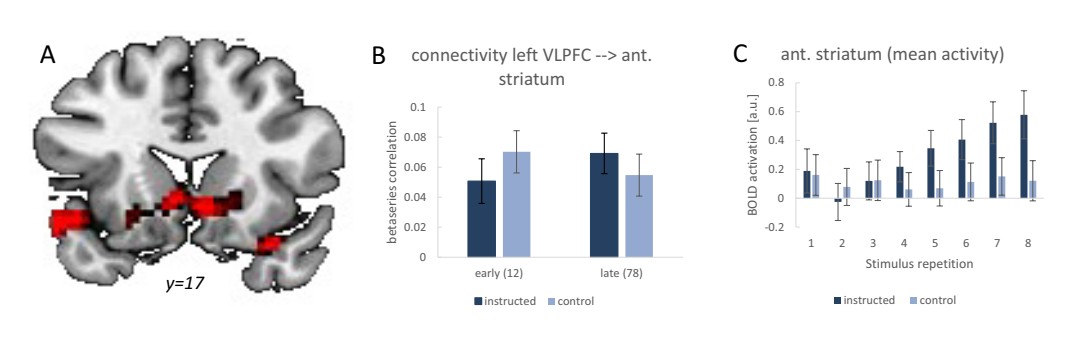

**Figure 8.** Summary of the functional connectivity analysis results for the left VLPFC seed region based on single-trial beta-series correlations. The analysis tested for a functional connectivity increase from early implementation trials (stimulus repetitions 1 and 2) to late implementation trials (stimulus repetitions 7 and 8) which was stronger for intentional learning blocks than control blocks. (**A**) Visualization of the significant effect in the anterior striatum. For display purposes the map shows voxels with p<0.001 uncorrected. (**B**) The detailed connectivity pattern for the anterior striatum cluster. (**C**) Mean activations at each stimulus repetition level based on a conventional univariate analysis for the anterior striatum cluster. Error bars represent 90% confidence intervals.

completely novel S-R rules and continued to be detectable throughout the first few implementation trials. This effect was highly specific for the VLPFC and it was virtually absent in the DLPFC. Importantly, we could show that identity-specific VLPFC pattern similarity effects were indeed preferentially observed in the intentional learning condition involving newly instructed S-R rules. This contrasted with the virtual absence of VLPFC pattern similarity effects in the control condition involving identical contingencies between the same word stimuli and responses yet without the need to memorize these contingencies for current or future task performance. This suggests that the VLPFC pattern similarity effect is unrelated to stimulus identity or response identity alone. Moreover, since novel word stimuli were predictive of the cued responses also in the control condition, incidental S-R learning (*Kelly, 2012*) might have occurred. If this assumption is made, we can additionally conclude that the VLPFC pattern similarity in the intentional learning condition is unlikely to reflect incidental S-R learning. Based on the current study design alone we cannot directly test for the presence of incidental S-R learning in the control condition. However, earlier behavioral results suggested that incidental S-R learning occurred in a condition comparable to the current control condition (*Frimmel et al., 2016*). Specifically, it was shown that response times significantly decreased across repeated implementation trials when novel stimuli were consistently paired with the same cued responses (as in the present control condition) compared to a condition in which novel stimuli were randomly assigned to the cued responses across repeated implementation trials.

The continued presence of VLPFC pattern similarity effects across the entire implementation phase was paralleled by increasing functional coupling between the VLPFC and the anterior striatum. This seems to suggest that the more and more fluent application of newly established rule representations is increasingly channelled through inter-regional cooperation during an early phase of short-term task automatization (cf., *Chein and Schneider, 2012*; *Ruge and Wolfensteller, 2016*). Interestingly, the continued representation of rule identity information within the VLPFC stands in stark contrast to the rapidly decreasing mean activity revealed by the univariate analysis. Moreover, while the multivariate pattern similarity effect was tightly confined to the VLPFC, the decreasing mean activity spread across VLPFC and DLPFC. This re-emphasizes the insight that mean activity results are unsuited to draw meaningful conclusions regarding the representational content of brain regions (*Coutanche, 2013*) and puts into perspective somewhat over-interpreted univariate analysis results we have reported earlier (*Ruge and Wolfensteller, 2010*).

The continued presence of rule-specific VLPFC pattern similarity effects referred to consecutive, non-overlapping repetition pairs (repetition pairs 1/2 and 3/4, plus 5/6 and 7/8 in experiment 2). Importantly, this does not automatically imply 'higher-order' pattern similarity in the sense that, for instance, representations identified in the beginning (repetition pair 1/2) are the same as those identified in the end (repetition pair 7/8). Evidence for higher-order pattern similarity would require testing for consistency of representations *across* the a-priori defined repetition pairs (e.g., combining

repetitions 1 and 4 or 1 and 8). This would however, require a different type of controlled trial sequence generation prior to data collection to ensure unbiased MVPA results. Furthermore, for similar reasons, the present study design is unsuited to track the representational transition from rule instruction *prior to* implementation and the first implementation of newly instructed rules. Such questions are of relevance, for instance, in order to test assumptions of an intuitive working memory interpretation of our VLPFC pattern similarity effects. One assumption could be that an abstract or symbolic working memory representation of the instructed S-R rules is initially formed during the instruction phase, and *the same* representation might then be maintained across the subsequent implementation phase in order to guide performance. Clearly, an empirical test of this type of representational stability predictions would require the analysis of pattern similarities with respect to the transition between instruction phase and implementation phase and regarding 'higher-order similarities' across consecutive implementation stages.

The immediate and continued presence of VLPFC rule representations during task implementation in the present study is distinctly different from observations reported by earlier electrophysiological studies in non-human primates in the context of trial-and-error learning (*Cromer et al., 2011*; *Pasupathy and Miller, 2005*). Those studies found that successful rule acquisition occurred (often quite abruptly) a few trials *before* rule-specific neural coding could be detected in the LPFC. In other words, even though overt behavior clearly suggested that a novel S-R rule had been successfully acquired, the lateral PFC did not seem to initially represent this rule. By contrast, anterior caudate neurons directly reflected improvements in behavioral accuracy (*Pasupathy and Miller, 2005*). This suggests that under trial-and-error learning conditions and in non-human primates the anterior caudate rather than the lateral PFC might be the place where novel task rules are initially represented. Further research is necessary to clarify whether these differences in representational dynamics are due to (i) differences between trial-and-error learning and instruction-based learning, (ii) general differences between species, or (iii) regional differences between the VLPFC region (area BA 44/45) identified in the present study and the DLPFC region (area BA 9/46) selectively examined in the electrophysiological studies.

## Representations of newly instructed rules and familiar rules

Our primary aim to track the initial representational dynamics of newly instructed task rules naturally required an 'aggregation-free' MVPA approach based on single-trial estimates associated with the trial-by-trial coding of individual S-R rule identities. This contrasts with earlier MVPA studies which relied in one way or the other on aggregation schemes either across time (*Bengtsson et al., 2009*; *Howard et al., 2015*; *Pauli et al., 2019*; *Schuck et al., 2015*; *Soon et al., 2008*; *Woolgar et al., 2011*) or across individual rule identities (*Braem et al., 2017*; *Cole et al., 2016*; *Cole et al., 2013*; *Kahnt et al., 2011*; *Muhle-Karbe et al., 2017*). Aggregation across individual rule identities improves signal-to-noise ratio regarding representations of task features on a more abstract level, but this generalization comes at the cost of losing specificity regarding individual rule identities. Similarly, aggregation across time, which typically involves aggregation across a large number of trials per rule identity, improves signal-to-noise ratio regarding each rule identity. Hence, this approach is obviously unsuited to track rapidly evolving representational dynamics spanning only a few trials. Instead, it is suited to examine representations involving well-familiarized task rules or to track slow learning processes evolving across blocks of large numbers of trials per rule identity.

Such aggregation-based studies could demonstrate that information regarding well-familiarized rule identities is flexibly represented within the prefrontal cortex under conditions that often require the prioritized implementation of one currently relevant task over competing alternative tasks (*Bengtsson et al., 2009*; *Bode and Haynes, 2009*; *Cole et al., 2011*; *Cole et al., 2016*; *Jackson and Woolgar, 2018*; *Schuck et al., 2015*; *Soon et al., 2008*; *Woolgar et al., 2015*; *Woolgar et al., 2011*; *Zhang et al., 2013*). This is consistent with similar findings reported in electrophysiological studies in non-human primates (*Asaad et al., 2000*; *Miller and Cohen, 2001*; *Roy et al., 2010*; *Stokes et al., 2013*). Overall. these studies nicely show that the prefrontal cortex flexibly codes anything of current task relevance, including information regarding task-relevant stimuli, responses, perceptual and conceptual categories, and transformation rules like those required in typical stimulus-response tasks (*Crittenden et al., 2016*; *Duncan, 2010*; *Fedorenko et al., 2013*; *Woolgar et al., 2016*). However, unlike the present study, these earlier conclusions were restricted

to already well-familiarized task features, and could hence not tell whether prefrontal representational flexibility also extends to completely novel tasks.

A number of pioneering MVPA studies specifically focusing on instruction-based learning could show that representations of *familiar* task features can be retrieved and re-cycled in the service of newly instructed tasks. One such study by *Muhle-Karbe et al. (2017)* identified LPFC activity patterns associated with highly familiar categorization routines regarding house pictures vs. face pictures – but, importantly, not regarding the concrete stimulus-response rules (e.g., the instructed responses for each of two different faces) underlying a multitude of individual face or house categorization tasks each involving a unique set of stimuli. Similar conclusions apply to related studies (*González-García et al., 2017*; *Palenciano et al., 2019*). Another approach pursued by Cole et al. (*Cole et al., 2011*; *Cole et al., 2013*) provided evidence for the principle of 'rule compositionality' (see also *Reverberi et al., 2012*; *Yang et al., 2019*). They showed that distributed activity and connectivity patterns of familiar task elements (e.g. same/different judgement or semantic categorization) were re-activated when these task elements were later combined with each other to construct a multitude of novel tasks defined by their specific combination and applied to a set of novel stimuli. Importantly, MVPA was based on aggregation over all those novel task compositions that shared one specific rule element. Hence, while being highly informative regarding the question of rule compositionality, this type of study does not speak to the question of how the identities of individual novel task compositions might be represented in the brain. This is exactly the question that was answered by the present study. Specifically, we identified the representations of novel combinations of familiar task elements (nouns, button presses).

## Memory mechanisms, task relevance, novelty, and causality

What type of memory mechanisms might be responsible for re-instantiating identity-specific VLPFC representations from one implementation trial to the next? As discussed further above, working memory maintenance could be an intuitive and parsimonious account of our results, but the current study was not designed to directly test relevant assumptions regarding working memory maintenance. In a similar vein, the present study was also not designed to delineate *proactive* working memory maintenance processes (preparing rule information for its future application) from *retroactive* episodic memory retrieval processes. Specifically, VLPFC representations might alternatively originate from the backwards-oriented retrieval of past episodes comprising contextual information experienced in spatiotemporal proximity with the current stimulus.

In this section, we will nevertheless elaborate on how our results could help to constrain the specific properties of such episodic retrieval processes. At first sight, an episodic retrieval account is difficult to reconcile with a significant VLPFC pattern similarity effect already in the earliest implementation stage 1 (involving implementation trials 1 and 2) and its unchanged magnitude across subsequent implementation stages. The reason is that the stimulus-triggered retrieval of past episodes would be quite different for implementation trials 1 and 2, hence resulting in a weak pattern similarity effect in stage 1. Specifically, the episode retrieved in implementation trial one comprises the past episodic context of a stimulus presented during the instruction phase (i.e. without behavioral implementation and generally within a distinctly different phase of the experiment). This contrasts with the past episode retrieved in implementation trial 2, which comprises the episodic context of the same stimulus during implementation trial 1 (i.e. including behavioral implementation within the same phase of the experiment). Subsequent implementation stages (e.g., stage two involving implementation trials 3 and 4) all involve already implemented S-R rules, which implies the retrieval of more similar episodic contexts. Hence, stimulus-specific pattern similarity effects should have been weak in stage 1, followed by a considerable increase in subsequent stages. This was not the case, however.

In light of these considerations, the episodic retrieval hypothesis can only be maintained under the 'assumption of consistency' that the VLPFC represents only those elements of a previously experienced stimulus-related episode that are consistently experienced across instruction phase and implementation phase alike. In theory, this would be the case if VLPFC representations were void of any reference to the instructed or implemented response and instead comprised information solely related to the perceptual and temporal context the stimulus appeared in (e.g., displayed on the computer screen, within a small room, within the past minute or so). Yet, it seems rather artificial to assume that VLPFC representations of past episodes would *exclude* of all things exactly the one

stimulus-related property that links the stimulus to the current task requirements, that is, the instructed response (for the notion of task relevance, see further below). If this argument is taken for granted, the consistency assumption can only be maintained (especially for stage 1) when VLPFC representations include stimulus-linked response information on an abstract or symbolic level (present in both the instruction phase and the implementation phase) while excluding episodic information regarding actual physical response implementation (which is absent during the instruction phase).

The episodic memory account in particular might stimulate questions as to whether the identified VLPFC representations are mere epiphenomenal reflections on the past while lacking any direct relevance for implementing the instructed response. On the one hand, as all correlative brain imaging approaches, this study clearly cannot provide evidence for a causal relevance of the identified VLPFC representations for the actual behavioral implementation of the instructed S-R rules. This would require a stimulation methodology like TMS as a means to directly manipulate VLPFC functioning – ideally in combination with an experimental design that allowed us to track within-trial activity dynamics to demonstrate that the engagement of item-specific VLPFC representations preceded actual response selection processes.

On the other hand, however, for the reasons elaborated above, it seems quite plausible to assume that VLPFC representations do comprise information regarding the instructed response for a given stimulus in an abstract or symbolic format. In this sense, the identified VLPFC representations are well qualified to serve an active 'task-set-like' role. Especially Experiment 2 provided further support for this notion suggesting that rule-specific VLPFC representations are found preferentially under conditions where the instructed links between word stimuli and responses are novel, arbitrary, and task-relevant. This was the case in the intentional learning condition of Experiment 2 (and all of Experiment 1) where correct responding required intact memory of novel and arbitrary links between word stimuli and manual responses as instructed during the preceding instruction phase. By contrast, this was different in the control condition of Experiment 2 where the correct response was directly specified by the spatial properties of the visual response cue (the vertical bars) presented in each and every implementation trial. Hence, correct responding could be based on non-arbitrary, spatially congruent links between visual cues and manual responses rather than the retrieval of arbitrary word-response links, which were in turn, not relevant for correct responding in the control condition (for the distinction between arbitrary and spatially-constrained visuomotor mapping, see *Toni et al., 2001*; *Wise and Murray, 2000*). Importantly, while spatially congruent cue-response relationships exploited during implementation trials of the control condition are in a sense 'instructions' too, novel learning demands are minimized. Our finding that VLPFC pattern similarity effects were virtually absent in the control condition suggests that 'instructed' S-R rules in terms of spatially congruent links between cues and responses are not encoded within the VLPFC. Moreover, this finding suggests that the mere repeated co-incidence of a spatially cued response and the concurrently displayed word stimulus in the control condition is not sufficient for the formation of rule-specific VLPFC representations. The likely reasons is that despite word-response links being arbitrary and novel also in the control condition, they do not bear any task relevance as their active memorization was not required for correct task performance – neither in the current trial nor for subsequent trials.

## Rule representations and the complexity of S-R instructions

An additional goal of experiment 1 was to explore the relationship between the strength or integrity of prefrontal rule representations and the extent of performance errors as a function of the complexity of S-R instructions. One of our original hypotheses was inspired by previous study results (*Cole et al., 2016*; *Rigotti et al., 2013*) and presumed that most errors would be committed due to damaged representations of the originally instructed S-R rules. Hence, a higher proportion of errors in the more difficult condition should be associated with a weaker identity-specific pattern similarity effect. However, if anything there was a non-significant trend towards a stronger identity-specific pattern similarity effect in the more difficult condition. A possible explanation of this null finding is based on a radically different account related to the notion of goal-neglect (*Bhandari and Duncan, 2014*; *Duncan et al., 2008*) and could explain why the strength or integrity of prefrontal cortex representations remained unaffected by differences in instructed rule complexity. Alluding to the difference between 'knowing' and 'doing' (*Demanet et al., 2016*; *Duncan et al., 2008*), more complex

instructions might induce more errors despite largely intact VLPFC representations. Instead, error rate might increase due to failures to correctly implement ('doing') correctly retrieved rules ('knowing'). This is consistent with VLPFC housing 'declarative' rather than 'procedural' rule representations (*Oberauer, 2009*) possibly related to the concept of an 'episodic buffer' within working memory (*Baddeley, 2000*; *Duncan et al., 2008*). Implementation errors despite 'knowing better' might occur when more complex instructions absorb additional control resources that are then lacking in order to prevent competing (e.g., perseverative) response tendencies from overriding the instructed correct response. Such a resource 'depletion' account would predict generally increased control effort following more complex instructions – including correctly performed trials. This prediction is indeed supported by the univariate analysis which revealed stronger mean activity in prefrontal cortex for more complex instruction blocks (paralleled by significantly increased response times). Additional support comes from the finding that response accuracies were positively associated with Raven's progressive matrices intelligence scores but not with simple working memory span. This seems to suggest that response errors were not so much related to the inability to memorize the instructions but rather to a more general cognitive control deficit reflected by the intelligence score. This is consistent with the observation that general intelligence is associated with goal neglect (*Duncan et al., 2008*).

## General conclusions

Our findings are suited to inform representational theories on how the prefrontal cortex supports behavioral flexibility. Specifically, we demonstrated that the VLPFC achieves flexibility not only by recycling established sub-routines in the service of novel task requirements but also by enabling the ad-hoc coding of novel task rules during early implementation trials right after their first-time instruction. This refutes alternative accounts that would have predicted an incremental process of rule formation in the prefrontal cortex possibly driven by leading signals generated by striatal areas. On the contrary, our findings suggest the reverse relationship between VLPFC and anterior striatum where the application of instantaneously available prefrontal rule representations seems to be increasingly channelled through inter-regional cooperation with the anterior striatum. Future research is needed however, to further clarify the relationship between striatal areas and prefrontal areas with respect to novel task learning under a greater variety of circumstances. In particular, this might include systematic explorations regarding (i) different types of intentional learning such as trial-and-error learning vs. instruction-based learning, (ii) different age groups or different species, and (iii) different time scales. Furthermore, future experimental work is required to (i) track the representational transition from rule instruction *prior to* implementation and the first implementation of newly instructed rules (ii) to better characterize the type of memory mechanisms that are responsible for re-instantiating identity-specific VLPFC representations from one trial to the next.

# Materials and methods

## Participants

The sample for experiment 1 consisted of 65 human participants (32 females, 33 males; mean age: 24.2 years, range 19–33 years). Three additional subjects could not be used due to incomplete data collection. Part of the present dataset was used in a previous methods-oriented paper (*Ruge et al., 2018b*). The sample for experiment 2 consisted of 70 human participants (39 female, 31 male; mean age: 23.9 years, range 19–33 years). Two additional subjects could not be used due to incomplete data collection. All participants were right-handed, neurologically healthy and had normal or corrected vision. The experimental protocol was approved by the Ethics Committee of the Technische Universität Dresden and conformed to the World Medical Association's Declaration of Helsinki. All participants gave written informed consent before taking part in the experiment and were paid 10 Euros per hour for their participation or received course credit.

## Tasks

Both experiments were based on modified versions of an established instruction-based learning paradigm (*Ruge and Wolfensteller, 2010*). Generally, the participants worked through a series of different novel tasks blocks. In each task block they were required to memorize novel task instructions

during an initial *instruction phase* during which response execution was not yet required. The instruction phase was followed by a manual *implementation phase* requiring task execution on a trial-by-trial basis by retrieving the previously encoded task rules from memory. In both experiments a task instruction comprised a set of novel stimulus-response (S-R) rule identities. The term 'rule identity' refers to a specific link between one unique stimulus and the response assigned to that stimulus. Each set of stimuli comprised either 4 or 10 written disyllabic German nouns which were mapped onto either 2 or 3 different manual button press responses (index, middle, or ring finger of the right hand). The number of responses was varied in order to encourage the memorization of all S-R rules and to avoid excessive use of short-cuts like 'these two stimuli require response A, hence all other stimuli require the other response' (*Liefooghe and De Houwer, 2018*). The number of task blocks requiring either 2 or 3 different responses was equally distributed across the different instruction conditions (easy/difficult in experiment 1 and intentional learning/control in experiment 2).

The start of an impending instruction phase was announced by the German word for 'memorize' ('Einprägen') displayed in red for 2 s, followed by the presentation of the first instructed noun. The start of the instruction phase announcement was delayed by a variable delay of 2 or 4 s relative to the start of a new measurement run or relative to the end of the preceding implementation phase. During instruction, the novel nouns were presented in rapid succession framed by two vertical bars to the left and to the right of the noun (see *Figure 1*). If a noun was closer to the left vertical bar, this indicated an index finger response. If a noun was closer to the right vertical bar, this indicated a ring finger response. If a noun was equally close to both vertical bars, this indicated a middle finger response. We only recruited right-handed subjects who were asked to use the right hand fingers for responding.

During the manual implementation phase which directly followed the instruction phase, the stimuli were presented in pseudo-random order such that each stimulus was presented 4 times (experiment 1) or 8 times (experiment 2). Each implementation phase was announced by the German word for 'implement' ('Ausführen') displayed in green for 2 s. There was no performance feedback after individual trials to avoid interference with reinforcement learning. The SOA varied randomly between 2 and 4 s in 0.5 s steps. The SOA interval was inserted *before* the start of a new trial to ensure that there was also random jitter between the end of the instruction phase and the beginning of the first implementation trial. After a variable delay of 2 or 4 s relative to the end of the last trial, the implementation phase ended with a display (2 s) of the mean performance accuracy computed across the preceding trials.

## Experiment 1 specifics

The aim of experiment 1 was twofold. First, we wanted to identify rule-specific neural representations with maximal statistical power and focused on the earliest phase of learning. We therefore realized a large number of 36 unique learning blocks each comprising only 4 repetitions of each of four stimuli. Second, we wanted to explore the relationship between the strength or integrity of prefrontal representations and the commission of performance errors. We therefore manipulated the complexity or difficulty of S-R instructions. The two difficulty conditions only differed regarding the number of instructed S-R rules (4 vs. 10) but not regarding the number of actually implemented S-R rules (always 4). In the difficult condition, 10 nouns were instructed and each was displayed for 1 s. In the easy condition, 4 nouns were instructed and each was displayed for 2 s. With respect to the subsequent implementation phase, the two conditions were identical, that is in either case, 4 nouns were presented. The subset of 4 out of 10 instructed nouns presented during the implementation phase of the difficult condition was selected such that 2 or 3 different responses were required equally often. Participants performed 18 blocks of each condition in pseudo-randomized order, which took approximately 40 min. Measurements were taken in three consecutive runs of ca. 13 min duration, each comprising 6 blocks of each difficulty condition. Also, the random delay before the start of each novel instruction phase and the delay before performance feedback was pseudo-randomized such that each SOA level occurred equally often for each difficulty condition. Before entering the scanner each participant completed a short practice session comprised of one novel task block for each difficulty condition with a separate stimulus set not used during the main experiment.

After completion of the instruction-based learning experiment in the scanner, participants performed a computerized simple digit span task to determine individual simple working memory span

scores (*Wechsler, 1997*). This score was chosen to obtain a relative pure measure of working memory storage in the absence of considerable executive control requirements. Random sequences of digits were displayed on the screen, one digit every second and each digit displayed only once within a sequence. Following a sequence, as many question marks as digits were displayed on the screen and subjects were required to reproduce the digits either in the forward or backward order. The first sequence started with three digits, followed by sequences of increasing number of digits (up to 10) if the previous answer was correct. If not, a new sequence with the same number of digits was displayed. If the answer was incorrect again, the test stopped. The final score was the maximal number of digits that was answered correctly.

Finally, participants performed a computerized short version of the standard progressive matrices intelligence test using only the two most difficult matrix sets (D and E) out of all five sets (*Raven, 2003*). Each set comprised 12 matrices presented in progressively difficult order. The non-standardized intelligence score was the sum of correctly solved matrices.

## Experiment 2 specifics
Experiment 2 was designed as a follow-up to experiment 1 to specifically test the hypothesis that prefrontal cortex representations can be preferentially identified for intentionally learned newly instructed S-R rules and that these representations are not merely related to the identities of the involved stimuli or responses. Therefore, experiment 2 included the easy condition only (i.e. 4 instructed and implemented S-R rules per task) and two types of conditions were realized, including an intentional learning condition (as in experiment 1) and a control condition. Different from the learning condition, in the control condition the instruction cues (the vertical bars) were omitted during the instruction phase but were instead presented together with the nouns during the implementation phase (see *Figure 2*). Hence, in the control condition no S-R rules could be memorized during the instruction phase and task implementation could rely entirely on the explicit response cues rather than memorized instructions. Additionally, experiment 2 was designed to track the representational dynamics across a more extended practice period. Therefore, each noun was presented 8 times during the implementation phase (instead of 4 times in experiment 1). Measurements were taken in three consecutive runs (18 min each) comprising 4 blocks of each condition (intentional learning and control) in pseudo-randomized order, amounting to a total of 12 blocks per condition (total duration approximately 54 min). Before entering the scanner each participant completed a short practice session comprised of one task block for each condition with a separate stimulus set not used during the main experiment. Different from experiment 1, measures of working memory span and general intelligence were not taken.

## Behavioral data analysis
Behavioral performance was assessed regarding mean response times for correct responses (RTs) and regarding response accuracies (proportion of correct responses). Mean RTs and response accuracies were each analyzed with repeated measures ANOVAs. In experiment 1, response accuracies were especially relevant as a measure of representational integrity which was targeted by the manipulation of instruction difficulty. Since there was no feedback provided after response execution, representational integrity might be quantified inadequately if accuracy was measured in 'objective' terms with reference to the originally instructed response. The reason is that - in case the originally instructed response is not properly recalled – participants might generate subjectively defined rule representations based on the response that was actually executed for a specific stimulus irrespective of whether this was the originally instructed response. To account for this, response accuracies were defined relative to the response that was executed upon the preceding occurrence of a specific stimulus. Since there is by definition no response execution prior to stimulus repetition 1, accuracy was in this case naturally defined relative to the instructed response, thus providing an 'objective' accuracy measure. This definition of response accuracies was applied in both experiments.

## Imaging methods
### Data acquisition
MRI data were acquired on a Siemens 3T whole body Trio System (Erlangen, Germany) with a 32 channel head coil. Ear plugs dampened scanner noise. After the experimental session structural

images were acquired using a T1-weighted sequence (TR = 1900 ms, TE = 2.26 ms, TI = 900 ms, flip = 9°) with a resolution of 1 mm x 1 mm x1 mm. Functional images were acquired using a gradient echo planar sequence (TR = 2000 ms, TE = 30 ms, flip angle = 80°). Each volume contained 32 slices that were measured in ascending order. The voxel size was 4 mm x 4 mm x 4 mm (gap: 20%). In addition, field maps were acquired with the same spatial resolution as the functional images in order to correct for inhomogeneity in the static magnetic field (TR = 352 ms, short TE = 5.32 ms, long TE = 7.78 ms, flip angle = 40°). The experiment was controlled by E-Prime 2.0.

## Preprocessing

The acquired fMRI data were analyzed using SPM12 running on MATLAB R2016a. First, the functional images were slice-time corrected, spatially realigned and unwarped using the acquired field maps. Each participant's structural image was co-registered to the mean functional image and segmented. Spatial normalization to MNI space was performed by applying the deformation fields generated by the segmentation process to the functional images (resolution: 3 mm x 3 mm x 3 mm). The images were not additionally smoothed prior to GLM estimation in order to suit the planned MVPA (*Kriegeskorte et al., 2006*). Instead each subjects' images were smoothed with 6 mm FWHM after the MVPA was completed.

## Voxelwise single-trial BOLD estimation

Voxel-wise BOLD activation was estimated based on the General Linear Model (GLM) approach implemented within the SPM12 framework using a first-order auto-regressive model and including a 1/128 Hz high-pass filter in experiment 1 and a 1/256 Hz high-pass filter in experiment 2 in order to accommodate different learning block lengths. During GLM estimation SPM's implicit analysis threshold was switched off and instead all non-brain voxels were masked out using SPM's intracerebral volume mask 'mask_ICV.nii'. This procedure was chosen to enable group level statistics for regions affected by susceptibility-induced signal loss in a few subjects.

BOLD activations during the implementation phase were modeled by using single-trial GLMs. We used the least-squares-separate (LSS) model approach (*Mumford et al., 2014*; *Mumford et al., 2012*) which included one regressor modeling one specific implementation trial and another regressor modelling all other implementation trials. To obtain estimates for each single implementation trial, we estimated as many different LSS models as there were trials. While LSS modeling is computationally much more time consuming, it has been argued to produce more robust estimates than other approaches (*Mumford et al., 2014*; *Mumford et al., 2012*). Regressors were created by convolving stick functions synchronized to stimulus onset with the SPM12 default canonical HRF. In experiment 1, this implies a total of 192 independent GLMs for each single-trial regressor per run (16 trials per task block times 12 task blocks), which amounts to 576 independent GLMs across all three runs. In experiment 2, this implies a total of 256 independent GLMs for each single-trial regressor per run (32 trials per task block times eight task blocks), which amounts to 768 independent GLMs across all three runs.

In addition to the single trial regressors modeling activity in the implementation phase, we included for each independent GLM additional regressors modeling activity associated with the instruction phase and with performance feedback at the end of each implementation phase, plus one regressor capturing constant activity level per measurement run. To appropriately capture BOLD activation during the instruction phase, spanning either 12 s (easy condition) or 14 s (difficult condition), we used Fourier basis set regressors including 20 different sine-wave regressors spanning 44 s which were time-locked to the onset of the start of the instruction phase. Using a Fourier basis set has the advantage to flexibly model any BOLD response shape associated with the extended instruction phase without making prior shape assumptions. An advantage over FIR modeling is that a Fourier basis set easily operates at micro-time resolution (SPM default TR/16) whereas FIR operates at TR resolution only (*Henson and Friston, 2007*). Performance feedback was modeled with a standard event-related HRF function time-locked to the onset of the feedback screen.

## Multivariate pattern analysis

The MVPA was based on single-trial beta estimates obtained for the implementation phase. Rule identity-specific activation patterns were determined by adopting a modified versions of the multi-

voxel pattern correlation approach (*Haxby et al., 2001*; *Kriegeskorte et al., 2008*) geared towards the unbiased computation of time-resolved identity-specific pattern correlations within runs. Specifically, identity-specific patterns were identified by computing the mean difference between (i) pattern correlations for re-occurrences of same stimuli and (ii) pattern correlations for occurrences of different stimuli. Such mean difference values were computed separately for each task block and within each task block separately for each successive implementation stage defined by two consecutive occurrences per stimulus (*Figure 3*). This procedure allowed us to analyze two implementation stages in experiment 1 (stage 1: stimulus repetitions 1 and 2; stage 2: stimulus repetitions 3 and 4). In experiment 2, two additional implementation stages could be analyzed involving stimulus repetitions 5 and 6 (stage 3) and stimulus repetitions 7 and 8 (stage 4). Finally, for each subject, the resulting mean difference values were averaged across task blocks separately for each implementation stage before being submitted to group-level statistical evaluation.

Importantly, recent work has highlighted that multi-voxel pattern analysis (MVPA) can be severely biased when BOLD response estimation involves systematic imbalance in model regressor correlations (*Mumford et al., 2014*; *Visser et al., 2016*). This problem occurs in situations where experimental conditions of interest are temporally dependent in the presence of overlapping BOLD activity. Obviously, this is the case in the current paradigm where consecutive implementation stages are inherently ordered and close in time. Based on simulations and a real data example we have recently shown that the sequence of stimulus occurrences can be constructed in a way that ensures unbiased multivariate results under conditions of overlapping single-trial BOLD responses within task blocks when consecutive implementation stages are of interest (*Ruge et al., 2018b*). For this technique to provide unbiased results it is crucial to generate appropriate stimulus sequences prior to data collection. Simply relying on randomized presentation of stimuli fails to ensure zero bias when the full random sequence is retrospectively divided into stimulus occurrences associated with different implementation stages. Instead, to ensure bias-free MVPA of consecutive implementation stages, stimulus sequence randomization needs to be done separately for each consecutive implementation stage within each task block. Specifically, for each task block the overall stimulus sequence was composed of 2 (experiment 1) or 4 (experiment 2) independently generated 'atomic' 8-trial sequences, each comprising two randomly distributed occurrences of each of the four nouns. On average across such atomic sequences, this approach guarantees unbiased MVPA due to the circumstance that non-zero bias regarding individual atomic sequences is distributed around zero mean (*Ruge et al., 2018b*). We furthermore took advantage of multiple novel task blocks per participant which allowed us to regress out bias-induced variance across blocks and thereby to obtain more robust results. Bias-induced variance regarding pattern similarity estimates was determined for subject-specific white-matter volumes (see below) with verified absence of significant multivariate effects on average across subjects (see *Figure 6B*).

The primary MVPAs were computed for regions-of-interest (ROIs) considering all voxels within a ROI simultaneously. Additional searchlight-based MVPAs were computed for the whole-brain volume with a spherical searchlight radius of 3 voxels (*Kriegeskorte et al., 2006*) as implemented in the CosmoMVPA toolbox (*Oosterhof et al., 2016*). The ROI-based approach was employed to be able to conveniently compare multivariate effects between different LPFC regions in a proper statistical way (*Nieuwenhuis et al., 2011*). The complementary whole-brain searchlight approach allowed us to also identify additional effects outside the pre-specified ROIs. In addition to the primary ROI-based MVPA which operated on the full voxel set within a ROI, a complementary ROI-based searchlight approach (three voxel radius) was used to localize MVPA effects within the anatomical ROIs with better spatial precision. The searchlight approach was chosen only as a complementary analysis step as the ROI-based MVPA operating on the full voxel set is clearly superior in terms of comparability across tests within and across experiments. Searchlight results were statistically evaluated at the peak-level with p<0.05, FWE-corrected for the whole-brain volume or for the ROI volume, respectively.

Four anatomically constrained ROIs were included based on the previous literature which had most consistently highlighted the potential relevance of lateral PFC regions (*Bengtsson et al., 2009*; *Cole et al., 2016*; *Cromer et al., 2011*; *Muhle-Karbe et al., 2017*; *Pasupathy and Miller, 2005*; *Woolgar et al., 2016*). Using the automatic anatomic labeling atlas (*Tzourio-Mazoyer et al., 2002*), we included for each hemisphere the ventrolateral PFC (according to the combined aal regions

'inferior frontal gyrus pars opercularis' and 'inferior frontal gyrus pars triangularis') and the dorsolateral PFC (aal region 'middle frontal gyrus').

The MVPA was based on all trials including correct trials and error trials alike. This allowed us to test how differences in the overall proportion of error trials would modulate the strength of identity-specific pattern similarity effects. This procedure was preferred over running separate MVPAs selectively based on either correct trials or error trials. Especially, the relatively small proportion of error trials in the easy instruction condition renders reliable pattern similarity estimates unfeasible.

## Univariate analysis of mean activity

The MVPA was complemented by a conventional univariate analysis computed for the MVPA ROIs. Instead of single-trial beta estimates, the univariate analysis was based on beta estimates collapsed across all trials per condition. In experiment 1 the conditions were defined by easy blocks vs. difficult blocks and by stimulus repetitions (1 to 4). In experiment 2 the conditions were defined by intentional learning blocks vs. control blocks and by stimulus repetitions (1 to 8). Error trials were excluded.

## Functional connectivity analysis

Functional connectivity changes were computed specifically for experiment 2 using the beta-series correlation approach based on the same single-trial estimates that were already generated for the MVPA (*Abdulrahman and Henson, 2016*; *Di et al., 2018*; *Rissman et al., 2004*). Error trials were excluded. Following-up on previous study results (*Ruge and Wolfensteller, 2013*; *Ruge and Wolfensteller, 2015*), we examined functional connectivity changes comparing late implementation trials (stimulus repetitions 7 and 8) with early implementation trials (stimulus repetitions 1 and 2) with a special focus on connectivity between the lateral PFC and the basal ganglia.

## Acknowledgements

This work was funded by the German Research Foundation (Deutsche Forschungsgemeinschaft, DFG), SFB940 A2 and Z2.

## Additional information

### Funding

| Funder | Grant reference number | Author |
| --- | --- | --- |
| Deutsche Forschungsgemeinschaft | SFB940 A2 | Hannes Ruge<br>Uta Wolfensteller |
| Deutsche Forschungsgemeinschaft | SFB940 Z2 | Hannes Ruge |

The funders had no role in study design, data collection and interpretation, or the decision to submit the work for publication.

### Author contributions

Hannes Ruge, Conceptualization, Formal analysis, Funding acquisition, Visualization, Methodology, Writing—original draft; Theo AJ Schäfer, Conceptualization, Methodology, Writing—review and editing; Katharina Zwosta, Conceptualization, Writing—review and editing; Holger Mohr, Methodology, Writing—review and editing; Uta Wolfensteller, Conceptualization, Funding acquisition, Writing—review and editing

### Author ORCIDs

Hannes Ruge https://orcid.org/0000-0001-9793-3859
Theo AJ Schäfer https://orcid.org/0000-0003-4102-559X

## Ethics

Human subjects: The experimental protocol was approved by the Ethics Committee of the Technische Universität Dresden (approval identifier: EK 545122015) and conformed to the World Medical Association's Declaration of Helsinki. All participants gave written informed consent before taking part in the experiment and were paid 10 Euros per hour for their participation or received course credit.

## Decision letter and Author response

Decision letter https://doi.org/10.7554/eLife.48293.sa1
Author response https://doi.org/10.7554/eLife.48293.sa2

## Additional files

### Supplementary files

• Transparent reporting form

### Data availability

Preprocessed single subject data and unthresholded whole-brain maps underlying the main results visualized in Figure 4, Figure 5, Figure 6, Figure 7, and Figure 8 are publicly available here: https://osf.io/vsbx8/.

The following dataset was generated:

| Author(s) | Year | Dataset title | Dataset URL | Database and Identifier |
|---|---|---|---|---|
| Ruge H, Schäfer TAJ, Zwosta K, Mohr H, Wolfensteller U | 2019 | Neural representation of individual rules after first-time instruction | https://osf.io/vsbx8/ | Open Science Framework, vsbx8 |

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
