## [Decision Letter]

**Acceptance summary:**

This study reports two experiments seeking to investigate the neural representations of newly instructed rules. Humans are able to receive instructions and then carry out new rules immediately. This ability to rapidly represent and carry out stimulus-response rules is a central aspect of human flexible behavior. However, little is known about how the brain represents newly instructed rules. The present fMRI study provides evidence that the ventrolateral prefrontal cortex codes newly instructed rules within the first two trials, that this coding is stable while univariate activity decreases, and that changes in VLPFC-striatal connectivity increases.

**Decision letter after peer review:**

Thank you for submitting your article "Initial neural representation of individual rules after first-time instruction" for consideration by *eLife*. Your article has been reviewed by three peer reviewers, and the evaluation has been overseen by David Badre as the Reviewing Editor and Timothy Behrens as the Senior Editor. The following individuals involved in review of your submission have agreed to reveal their identity: Apoorva Bhandari (Reviewer #1); Alexandra Woolgar (Reviewer #3).

The reviewers have discussed the reviews with one another and the Reviewing Editor has drafted this decision to help you prepare a revised submission.

The reviewers were in agreement that the topic of this set of experiments – the neural coding of newly instructed rules – is of high importance and potential impact. They also felt that the study was overall well-designed and carefully conducted. However, there were also several conclusions drawn in the paper that the reviewers felt were either poorly supported or would require additional analyses to support. We have consolidated these consensus concerns into the following list of essential revisions.

Essential revisions:

1) The most important claim of this paper is that neural representations of novel stimulus-response rules are established after instructions (or after the first implementation – see point 3 below). There is an alternative account of these findings that has not been ruled out, however. The pattern similarity effects may also reflect stimulus-specific episodic retrieval. With reference to the example trials shown in Figure 1, when subjects see the cue "water" on trial 3, they may be retrieving their episodic memory of the last time they saw the cue "water" and what they did in that context. Such a pattern of episodic retrieval would also predict a similarity structure consistent with that observed in the VLPFC in this study. Moreover, such episodic retrieval is unlikely in the control condition in Experiment 2 when the response is cued on every trial, thus predicting the absence of such a similarity structure. There is a deflationary version of this alternative that is problematic for the main claims here. Specifically, we know that people can retrieve items from LTM after one exposure to them. We also know that systems including VLPFC are important for this kind of retrieval. So, if rule implementation is distinct from this kind of episodic retrieval, then this would be problematic. Thus, it is important to either rule out this alternative, or discuss its implications. One way to potentially rule it out would be to show pattern similarity effects between different rules in the task that would only be predicted by the SR contingencies, but not by stimulus-based episodic retrieval.

2) Claims of the stability of the representation are overstated, unless they are supported with additional analysis. The pattern analysis approach employed in this study, though elegant, cannot support such a claim. This approach detects rule-specific, structured pattern similarity in sets of 2 trials (e.g. same-rule similarity > different-rule similarity across repetitions 1 and 2 or 3 and 4 in Experiment 1). While this provides evidence for rule-identity specific pattern coding across adjacent pairs, it does not provide evidence that the same rule identity-specific patterns are stably maintained across the first 4 (or first 8 in Experiment 2) trials. In theory, one way this issue could be addressed would be to analyze identity specific patterns in overlapping pairs of repetitions (1 and 2, 2 and 3, 3 and 4). Unfortunately, such an analysis is likely to be biased because of the way the trial sequences were constructed in order to enable analysis in each learning stage (i.e. using two, independent, 'atomic' 8-trial sequences). Therefore, without a control analysis (using realistic autocorrelated noise) or an alternative analysis approach that can provide more direct evidence, the claim of stable representations is not supported.

3) The primary claim that rule identity-specific representations are established immediately after instruction implies that something was tested about the representation of the rule once instructed but not yet implemented. This was not tested, however. Only data from implementation was analyzed. One reviewer further noted that to support the claim that the representation is established from the very first implementation then it is necessary to show that the identify-specific representations are above chance even for the 1 and 2 comparison specifically. This would establish that there is a consistency in representation between first and second implementation. This is as close as possible to establishing that it is present from the first use. The absence of an interaction with learning stage is not sufficient to establish a significant effect at all learning stages. Analysis of the instruction phase itself might also be helpful in establishing these early effects. In the absence of tests prior to implementation, then this claim should be revised.

4) The paper implies that explicitly learnt rules are contrasted with incidentally learnt ones to isolate processes associated with following explicit instructions. But, this is not what has been done. Instead, they contrast two conditions which both involve explicit instructions (previously learnt or currently given). Relatedly, since all analyses are for activity associated with rule implementation, and not learning per se, some rephrasing would help improve clarity. E.g. this was seen for implementation of previously instructed, and not currently instructed, rules?

5) The control for stimulus or response coding rather than rule coding largely comes from a lack of effect in the control condition. Beyond the potential weakness of arguing from a null, it is not clear why S-R rules would not be incidentally represented even though they are instructed on every trial in the control task. Thus, if one assumes that they are represented, then, in fact, the 'control task' results would suggest a lack of replication. Some discussion of this difference is merited.

6) An alternative explanation for the lack of effect of difficulty (ns trend towards increased rule coding under difficult conditions) is that there is an increased representation of the more difficult rules on the correct trials averaging out against poorer representation on error trials. There is precedence for this (Woolgar et al., 2016). To confirm which explanation applies, the analysis should be repeated using only correct trials. If the increase is still in the same direction or is stronger, then the 'knowing but not doing' explanation cannot hold, and the explanation should instead be an adaptive response in which more processing resources are allocated to more difficult tasks.

7) The searchlight analysis appears biased and can be dropped.

8) The pattern analysis approach used in the study is elegant, but it immediately brings to mind several concerns about the possibility of bias due to regressor correlation or temporal noise autocorrelation in fMRI. Reviewers had to carefully review the results in the referenced earlier paper (Ruge et al., 2018) in order to be satisfied that the results observed here are not due to bias. It will be very helpful to the reader if the argument about the unbiased nature of the method is recapitulated in summary in the current manuscript. This is particularly the case given the known difficulty with PFC MVPA – readers may be skeptical of the method without knowing some of the results from Ruge et al., 2018.

[Editors' note: further revisions were requested prior to acceptance, as described below.]

Thank you for re-submitting your article "Initial neural representation of individual rules after first-time instruction" for consideration by *eLife*. Your article has been re-reviewed by three peer reviewers, and the evaluation has been overseen by a Reviewing Editor and Timothy Behrens as the Senior Editor. The following individuals involved in review of your submission have agreed to reveal their identity: Apoorva Bhandari (Reviewer #1); Alexandra Woolgar (Reviewer #3).

The reviewers have now evaluated the revised manuscript with a focus on the points that were considered crucial on the previous round of review. Though the reviewers still believe that this work has the potential to make an important contribution to the literature, they did not find that some major concerns were addressed in your revision. After discussion, it was agreed that a further revision is merited. It is important to emphasize again that the reviewers are in consensus that these points are essential and must be addressed with revision to the language and strength of the claims.

First, the concern over the episodic retrieval account of the findings remains unaddressed, either through an analysis or through discussion. The reviewers remain concerned that this is an important alternative; they were not persuaded by the response. After discussion, there was agreement that this alternative needs to be addressed more directly. Ideally, it could be ruled out in the data with an additional control analysis. We are pessimistic that this will be possible, particularly as it may boil down to a question of whether this representation in VLPFC is causal rapid behavior; something that can't be addressed with fMRI. However, in the absence, of a new result that argues against episodic retrieval, this point must be addressed in the Discussion, with caveats as appropriate.

Second, the reviewers agreed that the paper continues to make claims that appear stronger than their empirical support.

- There is agreement that the study does not show rule coding prior to the first implementation of trials, so language implying that this is the case should be revised.

- Changing the definition of stability has not fully addressed this concern for reviewers. Again, the language should be revised to make this clear.

- The implicit (non-instructed) versus instructed distinction remains unclear for reviewers.

- Claims that the study looks at "novel combinations of elements without reference to the elements themselves" should be revised, as reviewers saw no evidence of this.

Finally, the reviewers have requested to see the ME of instruction type in VLPFC so that a representation of a rule can be distinguished from a stimulus.

---

## [Author Response]

Essential revisions:1) The most important claim of this paper is that neural representations of novel stimulus-response rules are established after instructions (or after the first implementation – see point 3 below). There is an alternative account of these findings that has not been ruled out, however. The pattern similarity effects may also reflect stimulus-specific episodic retrieval. With reference to the example trials shown in Figure 1, when subjects see the cue "water" on trial 3, they may be retrieving their episodic memory of the last time they saw the cue "water" and what they did in that context. Such a pattern of episodic retrieval would also predict a similarity structure consistent with that observed in the VLPFC in this study. Moreover, such episodic retrieval is unlikely in the control condition in Experiment 2 when the response is cued on every trial, thus predicting the absence of such a similarity structure. There is a deflationary version of this alternative that is problematic for the main claims here. Specifically, we know that people can retrieve items from LTM after one exposure to them. We also know that systems including VLPFC are important for this kind of retrieval. So, if rule implementation is distinct from this kind of episodic retrieval, then this would be problematic. Thus, it is important to either rule out this alternative, or discuss its implications. One way to potentially rule it out would be to show pattern similarity effects between different rules in the task that would only be predicted by the SR contingencies, but not by stimulus-based episodic retrieval.

We agree that the potential involvement of episodic long-term memory retrieval processes is an interesting issue bearing various ramifications. To start with, however, the current study was not specifically designed to directly address questions regarding specific memory processes. Rather, the primary goal was to identify rule-specific representations under conditions requiring *some sort of* memory retrieval in order to generate the correct response for a given stimulus (see also reply to point 4 below). In this general sense, correct performance was memory-dependent starting from implementation trial 1 (repetition 1) in Experiment 1 and in the instructed condition of Experiment 2, but not in the control condition of Experiment 2. The application of time-resolved MVPA enabled us to probe the presence of rule-specific representations at different stages during the implementation phase and thereby to rule out the specific concern expressed in the reviewers’ point 1 (and relatedly point 3). Specifically, in reviewers’ point 1 (and point 3 for that matter) it was argued that rule-specific representations might *not* yet be present in the first implementation trial after instruction according to the hypothesis that VLPFC representations might depend on the episodic retrieval of S-R links *implemented at least once* (i.e., only starting from repetition 2). If this ‘implementation-dependent’ episodic retrieval account was true, we should have been unable to detect rule-specific pattern similarities in the initial learning stage involving repetition 1 (implementation-dependent representations not yet established) and repetition 2 (representation established through episodic retrieval of the preceding implementation episode). Contrary to this prediction, additionally performed statistical tests (see subsection “MVPA (collapsed across experiments 1 and 2)”) explicitly demonstrate the presence of a VLPFC pattern similarity effect already in the initial learning stage (see also original Figure 5C). To re-iterate, such an effect can only be found if rule-specific activity patterns are elicited for both, repetition 1 and repetition 2.

In this context, however, we would like to emphasize that this finding does not generally refute the involvement of long-term memory retrieval processes (in addition to or instead of working memory processes). Specifically, it is well conceivable that long-term memory traces of the instructed S-R rules are established during the instruction phase, which are then retrieved starting from implementation trial 1 (i.e., independent of prior implementation). An interesting follow-up question would be whether long-term memory retrieval leads to or is based on VLPFC rule representations on implementation trial 1 (and following trials). An answer to this latter question could be constrained by testing for rule-specific pattern similarities involving data from the instruction phase (see also point 3). The present experiment was, however, not designed to do this in an unbiased way.

Regarding the distinction between working memory and long-term memory processes mentioned above, we indeed favour an episodic long-term memory retrieval account over a working memory account (see Discussion). This is mainly based on the repeated failure to find significant correlations between instruction-based learning performance and working memory scores – in the present paper as well as in earlier studies (Meiran et al., 2016; Ruge et al., 2018). Note, however, that this a rather speculative conclusions based on the absence of effects. In the instruction-based literature (see subsection “MVPA (experiment 2)”) there are proponents of both accounts favouring either WM (Cole, Braver, and Meiran, 2017) or episodic retrieval (Meiran, Liefooghe, and De Houwer, 2017; Ruge et al., 2018). Alternatively, one might speculate that VLPFC representations can originate from both working memory processes and episodic memory processes, depending on circumstances. For instance, it might be conceivable that in Experiment 1, the difficult condition exceeds WM capacity for most subjects and encourages a strategy preferentially relying on episodic memory whereas working memory maintenance might in turn be favoured by most subjects in the easy condition. In both cases, rule-specific VLPFC representations might be involved indistinguishably. Clearly, definite answers to these important questions require additional experimental work.

2) Claims of the stability of the representation are overstated, unless they are supported with additional analysis. The pattern analysis approach employed in this study, though elegant, cannot support such a claim. This approach detects rule-specific, structured pattern similarity in sets of 2 trials (e.g. same-rule similarity > different-rule similarity across repetitions 1 and 2 or 3 and 4 in Experiment 1). While this provides evidence for rule-identity specific pattern coding across adjacent pairs, it does not provide evidence that the same rule identity-specific patterns are stably maintained across the first 4 (or first 8 in Experiment 2) trials. In theory, one way this issue could be addressed would be to analyze identity specific patterns in overlapping pairs of repetitions (1 and 2, 2 and 3, 3 and 4). Unfortunately, such an analysis is likely to be biased because of the way the trial sequences were constructed in order to enable analysis in each learning stage (i.e. using two, independent, 'atomic' 8-trial sequences). Therefore, without a control analysis (using realistic autocorrelated noise) or an alternative analysis approach that can provide more direct evidence, the claim of stable representations is not supported.

This is a good point. It is indeed true, that due to the bias issue, stability of representations was defined by the continued presence of rule-specific activity patterns in consecutive, non-overlapping repetition pairs (repetition pairs 1/2 and 3/4, plus 5/6 and 7/8 in Experiment 2). In this sense we concluded that VLPFC represents rule identities across the entire implementation phase. However, we cannot conclude (and did not mean to) that, for instance, representations identified in the beginning (1/2) are the same as those identified in the end (7/8). Evidence for such ‘higher-order’ pattern similarity would require a test for consistency of representations *across* the a priori defined repetition pairs (e.g., combining repetitions 1 and 4 or 1 and 8). As the reviewers pointed out correctly, this is hard to achieve based on the present data set: Bias induced by sequentially dependent BOLD responses was only controlled for *within* the a priori defined repetition pairs.

In order to prevent misinterpretation of our results we added a paragraph at the start of the Discussion section now more precisely defining the term ‘stability’ used in the present paper: ‘Importantly, representational stability as defined here, refers to the continued presence of rule-specific pattern similarities regarding consecutive, non-overlapping repetition pairs (repetition pairs 1/2 and 3/4, plus 5/6 and 7/8 in Experiment 2). […] Such an analysis, however, would require a different type of controlled trial sequence generation prior to data collection to ensure unbiased MVPA results.’

3) The primary claim that rule identity-specific representations are established immediately after instruction implies that something was tested about the representation of the rule once instructed but not yet implemented. This was not tested, however. Only data from implementation was analyzed. One reviewer further noted that to support the claim that the representation is established from the very first implementation then it is necessary to show that the identify-specific representations are above chance even for the 1 and 2 comparison specifically. This would establish that there is a consistency in representation between first and second implementation. This is as close as possible to establishing that it is present from the first use. The absence of an interaction with learning stage is not sufficient to establish a significant effect at all learning stages. Analysis of the instruction phase itself might also be helpful in establishing these early effects. In the absence of tests prior to implementation, then this claim should be revised.

Please also see our reply to point 1 above. We provided the requested analyses regarding the initial learning stage (i.e. involving repetitions 1 and 2) in the subsection “MVPA (collapsed across experiments 1 and 2)” (see also Figure 5C): ‘Finally, to establish that an identity-specific pattern similarity effect was present in each learning stage two additional ANOVAs were computed each restricted to a single stage (i.e., early: repetitions 1/2 and late: repetitions 3/4). Specifically, for the early stage, there was a significant main effect region (F1,134=5.61; p(F)=.019; *ɳ_p_^2^* =.04) reflecting a significant effect for the VLPFC (F1,134=9.60; p(F)=.002; *ɳ_p_^2^* =.067) but not for the DLPFC (F1,134=1.78; p(F)=.184; *ɳ_p_^2^* =.013). For the late stage, there was again a significant main effect region (F1,134=9.41; p(F)=.003; *ɳ_p_^2^* =.066) reflecting a significant effect for the VLPFC (F1,134=7.53; p(F)=.007; *ɳ_p_^2^* =.053) but not for the DLPFC (F1,134=.001; p(F)=.974; *ɳ_p_^2^* <.001).’

MVPA analyses involving the instruction phase are hard to perform due to lack of bias control (see reply to point 1 above). Therefore, we cannot decide whether rule representations identified for the initial implementation stage (i.e. involving repetitions 1 and 2) were already present during the instruction phase. In fact, by using the word ‘after’ in our original title phrase ‘Initial neural representation of individual rules *after* first-time instruction’ we meant to imply that our conclusions refer to the subsequent implementation of instructed rules. This fact is now further highlighted at a prominent place in the General Conclusions section: ‘Furthermore, future experimental work is required to track the representational transition from rule instruction prior to implementation and the first implementation of newly instructed rules.’

4) The paper implies that explicitly learnt rules are contrasted with incidentally learnt ones to isolate processes associated with following explicit instructions. But, this is not what has been done. Instead, they contrast two conditions which both involve explicit instructions (previously learnt or currently given). Relatedly, since all analyses are for activity associated with rule implementation, and not learning per se, some rephrasing would help improve clarity. E.g. this was seen for implementation of previously instructed, and not currently instructed, rules?

We would like to point out that our concept of ‘instruction-based learning’ is essentially defined by the *necessity* to memorize instructions for subsequent implementation. This is the case if response cues are no longer presented during rule implementation. If the response cues are presented throughout the entire implementation phase as is the case in the control condition of Experiment 2, correct performance is perfectly possible without any memorization of the newly introduced S-R links. Importantly, in a recent behavioural study (Ruge et al., 2018) we have provided clear evidence that response cues presented during the first few novel rule implementation trials are processed very differently depending on whether subjects knew that these cues would be presented either indefinitely (incidental learning condition) or only for the first few implementation trials after which the correct responses needed to be retrieved from memory (intentional learning condition). Specifically, we found that, compared to the incidental learning condition, subjects spent additional effort for encoding the instructed S-R rules in the intentional learning condition (i.e., ‘instruction-based’ according to our conceptualization) and this encoding effort predicted subsequent memory-based performance. We therefore think it was fair to assume for the present study that the control condition of Experiment 2 likely involves incidental learning processes (if anything) rather than intentional learning processes which we hypothesized to be independent of higher-order control regions such as the lateral prefrontal cortex. Clearly, in retrospect, this assumption was confirmed by our results. We added a more detailed description of the Ruge, Karcz, et al., 2018 paper to further clarify this issue (Introduction and Results).

Finally, regarding terminology, in order to avoid confusion, we changed the term ‘learning stage’ into ‘implementation stage’ throughout the manuscript.

5) The control for stimulus or response coding rather than rule coding largely comes from a lack of effect in the control condition. Beyond the potential weakness of arguing from a null, it is not clear why S-R rules would not be incidentally represented even though they are instructed on every trial in the control task. Thus, if one assumes that they are represented, then, in fact, the 'control task' results would suggest a lack of replication. Some discussion of this difference is merited.

We would like to emphasize that our hypothesis was in fact not about a null effect in the control condition. Rather we hypothesized a stronger effect in the instructed learning condition compared to the control condition. We can see that our wording was a bit misleading in some places. In particular we repeatedly stated that the MVPA effect was ‚confined to the instructed learning condition‘. To clarify this, we now modified our wording to emphasize the nature of our hypothesis in terms of a differential effect greater than zero. In the Abstract we changed ‘confined to’ into ‘found preferentially for’. It now reads: ‘This was found preferentially for instructed stimulus-response learning in contrast to incidental learning involving identical contingencies between stimuli and responses’. Similarly, in the Introduction we changed ‘confined to intentional learning’ into ‘can be identified preferentially for’. It now reads: ‘…hypothesis that prefrontal cortex representations can be identified preferentially for intentional learning conditions involving instructed stimulus-response rules as compared to an incidental learning control condition involving the same contingencies between stimuli and responses but without the necessity to memorize these contingencies for correct performance.’ Similar changes were made in the Materials and methods section and in the Discussion.

Regarding the other issue (see also our response to point 4 above), we would like to point out that our hypothesis was based on our previous behavioral findings strongly suggesting that novel task rules are ‘intentionally’ encoded only when it is objectively necessary to memorize these rules for correct performance later on (Ruge, Karcz, et al., 2018) which is not the case in the control condition of Experiment 2. Hence, we agree with the reviewer’s point that, if anything, only incidental S-R learning takes place in the control condition of Experiment 2. Importantly, however, we hypothesized that incidental learning should be independent of higher-order control regions like the lateral prefrontal cortex. In this sense, we would strongly argue that the absence of a significant VLPFC effect in the control condition is not a ‘lack of replication’ but rather confirmation of our hypothesis. As mentioned in response to point 4 above, we added a more detailed description of the Ruge, Karcz, et al., 2018 paper to further clarify this issue (Introduction and Results).

6) An alternative explanation for the lack of effect of difficulty (ns trend towards increased rule coding under difficult conditions) is that there is an increased representation of the more difficult rules on the correct trials averaging out against poorer representation on error trials. There is precedence for this (Woolgar et al., 2016). To confirm which explanation applies, the analysis should be repeated using only correct trials. If the increase is still in the same direction or is stronger, then the 'knowing but not doing' explanation cannot hold, and the explanation should instead be an adaptive response in which more processing resources are allocated to more difficult tasks.

This is a good point. We repeated the ROI-based MVPA including correct trials only. However, the results remained qualitatively the same. If anything the original non-significant trend towards a stronger effect in the difficult condition became weaker (F1,64=1.12; p(F)=.294; *ɳ_p_^2^* =.017). The revised manuscript now reads ‘This trend towards stronger identity-specific rule representation in difficult blocks might hint towards a sharper representation of rules specifically in correctly performed difficult trials compared to correctly performed easy trials which might have been blurred by inclusion of erroneous trials (cf., Woolgar et al., 2016). […] However, the results remained qualitatively the same. If anything the trend towards stronger effects in the difficult condition became weaker (F1,64=1.12; p(F)=.294; *ɳ_p_^2^* =.017).’

7) The searchlight analysis appears biased and can be dropped.

We are unsure which searchlight analysis this point refers to. We performed two types of searchlight analyses.

1) We performed a whole-brain searchlight analysis. This was done in order not to miss potential MVPA effects in brain regions outside the preselected VLPFC and DLPFC ROIs. We do not see any bias there. Furthermore, this whole-brain analysis enabled us to demonstrate a significant VLPFC effect even after correction for the whole-brain volume. We think this finding is worth being reported (though not strictly necessary) as it can disperse potential criticism that we only used the ROI approach because we did not find significant results using a less constrained approach.

2) We performed searchlight analyses within preselected ROIs. While this analysis is certainly redundant in the sense that the primary ROI-based MVPA analysis (operating on the full set of voxels within each ROI simultaneously) already demonstrates the relevant pattern similarity effects, we disagree that it is biased in a relevant sense. The sole purpose of this analysis was to provide complementary information regarding the spatial ‘hot-spots’ of pattern similarity effects within the ROI (i.e., spatially more detailed information). The ROI-based searchlight approach was chosen only as a complementary analysis step as the ROI-based MVPA operating on the full voxel set is clearly superior in terms of comparability across tests within and across experiments (see modified text in the Materials and methods section). Importantly, this is unlike double-dipping in terms of generating false positive results which are not detectable otherwise. The point is that significance of the VLPFC pattern similarity effects had been established beforehand using the primary ROI-based MVPA analysis operating on the full voxel set. We would therefore prefer to include the searchlight part unless reviewers and/or Editor will insist on dropping it.

8) The pattern analysis approach used in the study is elegant, but it immediately brings to mind several concerns about the possibility of bias due to regressor correlation or temporal noise autocorrelation in fMRI. Reviewers had to carefully review the results in the referenced earlier paper (Ruge et al., 2018) in order to be satisfied that the results observed here are not due to bias. It will be very helpful to the reader if the argument about the unbiased nature of the method is recapitulated in summary in the current manuscript. This is particularly the case given the known difficulty with PFC MVPA – readers may be skeptical of the method without knowing some of the results from Ruge et al., 2018.

Thanks for pointing this out, we gladly adhere to this suggestion and have added a summary in the Materials and methods section and refer to this summary already in the Introduction in the revised version of the manuscript.

References:

Meiran, N., Pereg, M., Givon, E., Danieli, G., and Shahar, N. (2016). The role of working memory in rapid instructed task learning and intention-based reflexivity: An individual differences examination. Neuropsychologia. doi:10.1016/j.neuropsychologia.2016.06.037

[Editors' note: further revisions were requested prior to acceptance, as described below.]The reviewers have now evaluated the revised manuscript with a focus on the points that were considered crucial on the previous round of review. Though the reviewers still believe that this work has the potential to make an important contribution to the literature, they did not find that some major concerns were addressed in your revision. After discussion, it was agreed that a further revision is merited. It is important to emphasize again that the reviewers are in consensus that these points are essential and must be addressed with revision to the language and strength of the claims.

First, we would like to thank the reviewers and the editor for their once again thorough and constructive comments. We concede that our previous revisions were in part a bit superficial and we thank the reviewers for insisting on further clarification. We hope our revision are now satisfactory. Please note that different from the last round, we now addressed all points not only in the reply letter but without exception also in the main manuscript. We hope this improves clarity even further.

First, the concern over the episodic retrieval account of the findings remains unaddressed, either through an analysis or through discussion. The reviewers remain concerned that this is an important alternative; they were not persuaded by the response. After discussion, there was agreement that this alternative needs to be addressed more directly. Ideally, it could be ruled out in the data with an additional control analysis. We are pessimistic that this will be possible, particularly as it may boil down to a question of whether this representation in VLPFC is causal rapid behavior; something that can't be addressed with fMRI. However, in the absence, of a new result that argues against episodic retrieval, this point must be addressed in the Discussion, with caveats as appropriate.

In light of the renewed criticism expressed in the reviewers’ comments, we can now see how our original reply fell short of addressing all aspects implied by the different reviewer comments regarding the episodic retrieval issue. We now provide a more comprehensive reply and we also added a new sub-section in the Discussion of the manuscript elaborating in full detail on these issues (” Memory mechanisms, task relevance, novelty, and causality”).

On a general level, we would like to stress that we were not at any point trying to imply that VLPFC representations could not be the result of episodic memory retrieval processes. We do argue, however, that it is rather implausible – though logically possible – to assume that such VLPFC representations of past episodes would *exclude* of all things exactly the one stimulus-related property that links the stimulus to the current task requirements, that is, the instructed response. In the re-revision we are now trying to do a better job carving out which specific assumptions must be made to be able to explain our observations in terms of the retrieval of past S-R episodes (see point 1 below).

Furthermore, we are now addressing a second question implied by the reviewer comments which we missed in our previous reply. Specifically, are VLPFC representations mere reflections on the past without any relevance for controlling action implementation in the present trial or do they instead play a role in controlling current task implementation in the sense of ‘task set’ representations? We agree that based on the current experiments we cannot provide evidence for a causal relevance of the identified VLPFC representations for actual behavioral implementation of an instructed S-R link. We do believe however, that the virtual absence of VLPFC pattern similarity effects in the control condition of Experiment 2 suggests that VLPFC represents S-R information only when it is novel and when it bears task relevance. Please see our detailed reply below (point 2).

1) Our original reply regarding the episodic retrieval hypothesis was based on the reasoning that this hypothesis is – at first sight – hard to reconcile with a significant pattern similarity effect in the earliest implementation stage 1 (involving implementation trials 1 and 2) and its unchanged magnitude across subsequent implementation stages. The reason is that the stimulus-triggered retrieval of past episodes would be quite different for implementation trials 1 and 2, hence resulting in a weak pattern similarity effect in stage 1. Specifically, the episode retrieved in implementation trial 1 comprises the past episodic context of a stimulus presented during the instruction phase (i.e. without behavioral implementation and generally within a distinctly different phase of the experiment). This contrasts with the past episode retrieved in implementation trial 2, which comprises the episodic context of the same stimulus during implementation trial 1 (i.e. including behavioral implementation within the same phase of the experiment). Subsequent implementation stages (e.g., stage 2 involving implementation trials 3 and 4) all involve already implemented S-R rules, which clearly implies retrieval of more similar episodic contexts. Hence, stimulus-specific pattern similarity effects should have been weak in stage 1, followed by a considerable increase in subsequent stages. This was not the case, however. We understand that this line of reasoning is similar to the view expressed by reviewer 3 stating that it seems ‘a more natural conclusion to think that the similarity [effect] would be driven mainly by the repeated implementation (which is the same each time) rather than a memory (which is different on the first trial).’

In light of these considerations, the episodic retrieval hypothesis can only be maintained under the ‘assumption of consistency’ that the VLPFC represents only those elements of a previously experienced stimulus-related episode that are consistently experienced across instruction phase and implementation phase alike. In theory, this would be the case if VLPFC representations were void of any reference to the instructed or implemented response and instead comprised information solely related to the perceptual or temporal context the stimulus appeared in (e.g., displayed on the computer screen, within a small room, within the past minute or so). Yet, it seems rather artificial to assume that VLPFC representations of past episodes would exclude of all things exactly the one stimulus-related property that links the stimulus to the current task requirements, that is, the instructed response (for the notion of task relevance, see further below). If this argument is taken for granted, the consistency assumption can only be maintained (especially for stage 1) when VLPFC representations include stimulus-linked response information on an abstract or symbolic level (present in both the instruction phase and the implementation phase) while excluding episodic information regarding actual physical response implementation (which is absent during the instruction phase).

This all said it is important to note that our results are equally consistent with a seemingly more parsimonious working memory account: Symbolic working memory representations of the instructed S-R links might be formed during the instruction phase and then be maintained across the implementation phase. Yet, the present study is unsuited to provide a definite answer regarding which memory mechanism better explains our observations. We have now added this line of reasoning in the context of the ‘stability of VLPFC representations’ issue discussed in the Discussion.

2) We understand the reviewer comments (reviewer 2 in particular) in the sense that is unclear whether the identified VLPFC representations are mere epiphenomenal reflections on the past (lacking any direct relevance for implementing the instructed response) or instead play a role in controlling current task implementation in the sense of ‘task set’ representations. We concede that based on the current design we cannot provide evidence for a causal relevance of the identified VLPFC representations for actual behavioral implementation of an instructed S-R link. This is a limitation common to all correlative imaging studies. One way to overcome this limitation would be to use a brain stimulation methodology like TMS as a means to directly manipulate VLPFC functioning – ideally in combination with an experimental design that allowed us to track within-trial activity dynamics to demonstrate that the activation of item-specific VLPFC representations preceded actual response selection processes.

Importantly however, for the reasons elaborated in point 1 above, we think it is plausible to assume that VLPFC representations do comprise information regarding the instructed response for a given stimulus in an abstract or symbolic format. In this sense, the identified VLPFC representations are well qualified to serve an active ‘task-set-like’ role. Especially Experiment 2 provided further support for this notion suggesting that rule-specific VLPFC representations are found preferentially under conditions where the instructed links between stimuli and responses are novel, arbitrary, and task-relevant. This was the case in the intentional learning condition of Experiment 2 (and all of Experiment 1) where correct responding required intact memory of novel and arbitrary links between word stimuli and manual responses as instructed during the preceding instruction phase. By contrast, this was different in the control condition of Experiment 2 where the correct response was directly specified by the spatial properties of the visual response cue (the vertical lines) presented in each and every implementation trial. Hence, correct responding could be based on non-arbitrary, spatially congruent links between visual cues and manual responses rather than the retrieval of arbitrary word-response links, which were in turn, not relevant for correct responding in the control condition (for the distinction between arbitrary and spatially-constrained visuomotor mapping, see Toni et al., 2001; Wise and Murray, 2000). Importantly, while spatially congruent cue-response relationships exploited during implementation trials of the control condition are in a sense ‘instructions’ too, novel learning demands are minimized. Our finding that VLPFC pattern similarity effects were virtually absent in the control condition suggests that ‘instructed’ S-R rules in terms of spatially congruent links between cues and responses are not encoded within the VLPFC. Moreover, this finding suggests that the mere repeated co-incidence of a spatially cued response and the concurrently displayed word stimulus in the control condition is not sufficient for the formation of rule-specific VLPFC representations. The likely reasons is that despite word-response links being arbitrary and novel also in the control condition, they do not bear any task relevance as their active memorization was not required for correct task performance – neither in the current trial nor for subsequent trials

Second, the reviewers agreed that the paper continues to make claims that appear stronger than their empirical support.- There is agreement that the study does not show rule coding prior to the first implementation of trials, so language implying that this is the case should be revised.

We have now changed the wording everywhere in the manuscript, starting with the Title.

Original Title: ‘Initial neural representation of individual rules after first-time instruction’

Revised Title: ‘Neural representation of newly instructed rule identities during early implementation trials’

Original Abstract: ‘…to uncover the elusive representational states following newly instructed arbitrary rules such as ‘for coffee, press red button’, while transitioning from ‘knowing what to do’ to ‘actually doing it’.’

Revised Abstract: ‘to uncover the elusive representational states during the first few implementations of arbitrary rules such as ‘for coffee, press red button’ following their first-time instruction.’

Original Abstract: ‘VLPFC representations were established right after first-time instruction and remained stable across early implementation trials.’

Revised Abstract: ‘Identity-specific representations were detectable starting from the first implementation trial and continued to be present across early implementation trials.’

Original Introduction:‘…how the concrete rules of newly instructed tasks are initially represented in the human PFC right after their first-time instruction.’

Revised Introduction:‘…how the concrete rules of newly instructed tasks are initially represented in the human PFC during early implementation trials right after their first-time instruction.’

Original Introduction:‘…to uncover the rapidly evolving representational dynamics following novel rule instructions’

Revised Introduction:‘…to uncover the rapidly evolving representational dynamics while implementing novel rule instructions for the first time’

- Changing the definition of stability has not fully addressed this concern for reviewers. Again, the language should be revised to make this clear.

We changed the wording throughout the manuscript according to the reviewers’ request (thanks to reviewer 1 for some quite useful suggestions). We hope that together with the operational definition of how representational dynamics were computed (Discussion), we have now satisfactorily resolved this issue.

Original Abstract: ‘VLPFC representations were established right after first-time instruction and remained stable across early implementation trials’

Revised Abstract: ‘Identity-specific representations were detectable starting from the first implementation trial and continued to be present across early implementation trials.’

Original Introduction:‘If so, the next question then regards the stability of these initially formed representations. One possibility is that initial representations are rapidly fading…’

Revised Introduction:‘If so, the next question then regards the continued presence of such type of representation. One possibility is that the initial presence of prefrontal rule representations is rapidly fading…’

Original Introduction:‘Alternatively, stable prefrontal rule representations might continue to be important for successful task implementation…’

Revised Introduction:‘Alternatively, the presence of prefrontal rule representations might continue to be important for successful task implementation…’

Similar changes were made in the Discussion. For instance, the Discussion it now reads: ‘Furthermore, the continued presence of VLPFC pattern similarity effects across the entire implementation phase was paralleled by…’

Furthermore, in the Discussion we now discuss this issue with reference to a concrete example: ‘The continued presence of rule-specific VLPFC pattern similarity effects referred to consecutive, non-overlapping repetition pairs (repetition pairs 1/2 and 3/4, plus 5/6 and 7/8 in Experiment 2). […] Clearly, an empirical test of this type of representational stability predictions would require the analysis of pattern similarities with respect to the transition between instruction phase and implementation phase and regarding ‘higher-order similarities’ across consecutive implementation stages.’

- The implicit (non-instructed) versus instructed distinction remains unclear for reviewers.

We concede that our previous revision introduced a contradiction into our line of reasoning. This originated from a rather one-sided response to the original reviewer comments, which we took as a request for solely strengthening our claim regarding the intentional character of the instruction-based learning condition in contrast to the control condition. To this end, we added two qualifying statements in the original revision. First, we highlighted more strongly than before that the control condition did not require intentional memorization of the S-R contingencies for correct performance in present or future trials. Second, we suggested that the control condition might not even involve incidental S-R learning. The first statement is certainly beyond doubt from a logical stance, and in addition, there is empirical support for the claim that subjects refrain from intentional S-R learning in the absence of obligatory memorization demands (Ruge, Karcz, et al., 2018). To improve clarity of this point even further, we now reformulated the related paragraph at the end the Introduction. The second statement, however, was unfortunate since it not only contradicted our claim that the control condition also serves as a control for incidental S-R learning (beyond stimulus identity or response identity alone) but it also contradicts earlier empirical evidence that we neglected in the previous revision (Frimmel et al., 2016). Nevertheless, the reviewers are right in noting that based on the *current* data we cannot decide whether incidental learning took place at all in the control condition. This would require a direct comparison between the current control condition and a condition with random pairings between stimuli and the cued responses. In Frimmel et al., 2016, we have reported the results of such a comparison, which strongly suggested that incidental S-R learning took place in a condition comparable to the current control condition. We therefore think that it is indeed justified to use the current control condition as a control for incidental S-R learning.

As a compromise – acknowledging a certain weakness relying on ‘external’ evidence from Frimmel et al. – we now de-emphasize the incidental S-R learning part by consistently using the term ‘control condition’ rather than ‘incidental learning condition’. Moreover, we no longer state that the control condition was used as a control for incidental S-R learning (even though we strongly believe it is). For instance, in the Abstract we no longer refer to ‘incidental learning’ but rather simply use the neutral wording ‘…preferentially for conditions requiring the memorization of instructed S-R rules for correct performance‘. We also removed contradictory the statement in the Introduction calling into question the presence of incidental S-R learning in the control condition.

The first and only time we now mention incidental S-R learning is in the Discussion: ‘This contrasted with the virtual absence of VLPFC pattern similarity effects in the control condition involving identical contingencies between the same word stimuli and responses yet without the need to memorize these contingencies for current or future task performance. […] Specifically, it was shown that response times significantly decreased across repeated implementation trials when novel stimuli were consistently paired with the same cued responses (as in the present control condition) compared to a condition in which novel stimuli were randomly assigned to the cued responses across repeated implementation trials.’

Regarding the term ‘incidental learning’ we think that it can be used to describe S-R learning in the control condition, following a definition from the Encyclopedia of the Sciences of Learning (Kelly, 2012): ‘Incidental learning refers to any learning that is unplanned or unintended. It develops while engaging in a task or activity and may also arise as a by-product of planned learning. “Incidental learning” can imply that the acquisition of knowledge is unconscious in nature, though in contrast to implicit learning, there is no expectation that such knowledge should remain largely inaccessible to conscious awareness. However, note that some articles may refer to implicit learning tasks as incidental without making the above distinction. There is also a suggestion, mainly from an educational perspective, that incidental learning involves subsequent conscious reflection on material that was consciously noted at time of study but not recognized as relevant or useful.’

As one final measure, in order to avoid confusion related to the terms ‘instructed intentional learning condition’ vs. ‘uninstructed control condition’ (since the control conditions also involves some sort of ‘instruction’, see related discussion subsection “Memory mechanisms, task relevance, novelty, and causality”), we now consistently use the labels ‘intentional learning condition’ vs. ‘control condition’. See also changes made to Figure 2.

- Claims that the study looks at "novel combinations of elements without reference to the elements themselves" should be revised, as reviewers saw no evidence of this.

We deleted the respective statements throughout the manuscript:

Original Abstract: ‘These findings inform representational theories on how the prefrontal cortex supports behavioural flexibility, specifically by enabling the ad-hoc coding of novel task rules without recourse to representations of familiar sub-routines.’

Revised Abstract: ‘These findings inform representational theories on how the prefrontal cortex supports behavioural flexibility, specifically by enabling the ad-hoc coding of newly instructed individual rule identities during their first time implementation.’

Original Introduction: ‘Yet, as of now, it remains unknown whether similar regions also code novel combinations of familiar task elements (here: nouns, button presses) without reference to representations of the individual task elements themselves.’

Revised Introduction: ‘Yet these studies were not designed to determine how the newly recomposed rule identities are individually represented in the brain.’

Similar changes were made in the Discussion and in the General Conclusions.

Finally, the reviewers have requested to see the ME of instruction type in VLPFC so that a representation of a rule can be distinguished from a stimulus.

We added the requested post-hoc tests: ‘Post-hoc tests showed that the main effect of instruction type was exclusively significant in the VLPFC both in the early implementation stage (F_1,69_=4.60; p(F)<.036; *ɳ_p_^2^* =.062) and in the late implementation stage (F_1,69_=4.05; p(F)<.048; *ɳ_p_^2^* =.055), but not in the DLPFC (early: F_1,69_=.46; p(F)=.50; *ɳ_p_^2^* =.007; late: F_1,69_=.012; p(F)<.91; *ɳ_p_^2^* <.0001).’